# DISENTANGLED GENERATIVE CAUSAL REPRESENTATION LEARNING

## ABSTRACT

This paper proposes a Disentangled gEnerative cAusal Representation (DEAR) learning method. Unlike existing disentanglement methods that enforce independence of the latent variables, we consider the general case where the underlying factors of interests can be causally correlated. We show that previous methods with independent priors fail to disentangle causally correlated factors. Motivated by this finding, we propose a new disentangled learning method called DEAR that enables causal controllable generation and causal representation learning. The key ingredient of this new formulation is to use a structural causal model (SCM) as the prior for a bidirectional generative model. The prior is then trained jointly with a generator and an encoder using a suitable GAN loss incorporated with supervision. Theoretical justification on the proposed formulation is provided, which guarantees disentangled causal representation learning under appropriate conditions. We conduct extensive experiments on both synthesized and real datasets to demonstrate the effectiveness of DEAR in causal controllable generation, and the benefits of the learned representations for downstream tasks in terms of sample efficiency and distributional robustness.

## 1 INTRODUCTION

Consider the observed data $x$ from a distribution $q_x$ on $\mathcal{X} \subseteq \mathbb{R}^d$ and the latent variable $z$ from a prior $p_z$ on $\mathcal{Z} \subseteq \mathbb{R}^k$. In bidirectional generative models (BGMs), we are normally interested in learning an *encoder* $E : \mathcal{X} \to \mathcal{Z}$ to infer latent variables and a *generator* $G : \mathcal{Z} \to \mathcal{X}$ to generate data, to achieve both representation learning and data generation. Classical BGMs include Variational Autoencoder (VAE) (Kingma & Welling, 2014) and BiGAN (Donahue et al., 2017). In representation learning, it was argued that an effective representation for downstream learning tasks should disentangle the underlying factors of variation (Bengio et al., 2013). In generation, it is highly desirable if one can control the semantic generative factors by aligning them with the latent variables such as in StyleGAN (Karras et al., 2019). Both goals can be achieved with the disentanglement of latent variable $z$, which informally means that each dimension of $z$ measures a distinct factor of variation in the data (Bengio et al., 2013).

Earlier unsupervised disentanglement methods mostly regularize the VAE objective to encourage independence of learned representations (Higgins et al., 2017; Burgess et al., 2017; Kim & Mnih, 2018; Chen et al., 2018; Kumar et al., 2018). Later, Locatello et al. (2019) show that unsupervised learning of disentangled representations is impossible: many existing unsupervised methods are actually brittle, requiring careful supervised hyperparameter tuning or implicit inductive biases. To promote identifiability, recent work resorts to various forms of supervision (Locatello et al., 2020b; Shu et al., 2020; Locatello et al., 2020a). In this work, we also incorporate supervision on the ground-truth factors in the form stated in Section 3.2.

Most of these existing methods are built on the assumption that the underlying factors of variation are mutually independent. However, in many real world cases the semantically meaningful factors of interests are not independent (Bengio et al., 2020). Instead, semantically meaningful high-level variables are often causally correlated, *i.e.*, connected by a causal graph. In this paper, we prove formally that methods with independent priors fail to disentangle causally correlated factors. Motivated by this observation, we propose a new method to learn disentangled generative causal representations called DEAR. The key ingredient of our formulation is a structured causal model (SCM) (Pearl et al.,

2000) as the prior for latent variables in a bidirectional generative model. With some background knowledge on the binary causal structure, the causal prior is then learned jointly with a generator and an encoder using a suitable GAN (Goodfellow et al., 2014) loss. We establish theoretical guarantees for DEAR to learn disentangled causal representations under appropriate conditions.

An immediate application of DEAR is causal controllable generation, which can generate data from any desired interventional distributions of the latent factors. Another useful application of disentangled representations is to use such representations in downstream tasks, leading to better sample complexity (Bengio et al., 2013; Schölkopf et al., 2012). Moreover, it is believed that causal disentanglement is invariant and thus robust under distribution shifts (Schölkopf, 2019; Arjovsky et al., 2019). In this paper, we demonstrate these conjectures in various downstream prediction tasks for the proposed DEAR method, which has theoretically guaranteed disentanglement property.

We summarize our main contributions as follows:

- We formally identify a problem with previous disentangled representation learning methods using the independent prior assumption, and prove that they fail to disentangle when the underlying factors of interests are causally correlated.
- We propose a new disentangled learning method, DEAR, which integrates an SCM prior into a bidirectional generative model, trained with a suitable GAN loss.
- We provide theoretical justification on the identifiability of the proposed formulation.
- Extensive experiments are conducted on both synthesized and real data to demonstrate the effectiveness of DEAR in causal controllable generation, and the benefits of the learned representations for downstream tasks in terms of sample efficiency and distributional robustness.

## 2 OTHER RELATED WORK

**GAN-based disentanglement methods.** Existing methods, including InfoGAN (Chen et al., 2016) and InfoGAN-CR (Lin et al., 2020), differ from our proposed formulation mainly in two folds. First they still assume an independent prior for latent variables, so suffer from the same problem with previous VAE-based methods mentioned above. Besides, the idea of InfoGAN-CR is to encourage each latent code to make changes that are easy to detect, which actually applies well only when the underlying factors are independent. Second, InfoGAN as a bidirectional generative modeling method further requires variational approximation apart from adversarial training, which is inferior to the principled formulation in BiGAN and AGES (Shen et al., 2020) that we adopt.

**Causality with generative models.** CausalGAN (Kocaoglu et al., 2018) and a concurrent work (Moraffah et al., 2020) of ours, are unidirectional generative models (*i.e.*, a generative model that learns a single mapping from the latent variable to data) that build upon a cGAN (Mirza & Osindero, 2014). They assign an SCM to the conditional attributes while leave the latent variables as independent Gaussian noises. The limit of a cGAN is that it always requires full supervision on attributes to apply conditional adversarial training. And the ground-truth factors are directly fed into the generator as the conditional attributes, without an extra effort to align the dimensions between the latent variables and the underlying factors, so their models have nothing to do with disentanglement learning. Moreover their unidirectional nature makes it impossible to learn representations. Besides they only consider binary factors whose consequent semantic interpolations appear non-smooth, as shown in Appendix D. CausalVAE (Yang et al., 2020) assigns the SCM directly on the latent variables, but built upon iVAE (Khemakhem et al., 2020), it adopts a conditional prior given the ground-truth factors so is also limited to fully supervised setting.

## 3 PROBLEM SETTING

### 3.1 GENERATIVE MODEL

We first describe the probabilistic framework of disentangled learning with supervision. We follow the commonly assumed two-step data generating process that first samples the underlying generative factors, and then conditional on those factors, generates the data (Kingma & Welling, 2014). During the generation process, the generator induces the generated conditional $p_G(x|z)$ and generated joint distribution $p_G(x, z) = p_z(z)p_G(x|z)$. During the inference process, the encoder induces the encoded conditional $q_E(z|x)$ which can be a factorized Gaussian and the encoded joint distribution

$q_E(x, z) = q_x(x)q_E(z|x)$. We consider the following objective for generative modeling:

$$L_{\text{gen}} = D_{\text{KL}}(q_E(x, z), p_G(x, z)), \tag{1}$$

which is shown to be equivalent to the evidence lower bound used in VAEs up to a constant, and allows a closed form only with factorized Gaussian prior, encoder and generator (Shen et al., 2020).

Since constraints on the latent space are required to enforce disentanglement, it is desired that the distribution family of $q_E(x, z)$ and $p_G(x, z)$ should be large enough, especially for complex data like images. Normally more general implicit distributions are favored over factorized Gaussians in terms of expressiveness (Karras et al., 2019; Mescheder et al., 2017). Then minimizing (1) requires adversarial training, as discussed detailedly in Section 4.3.

### 3.2 SUPERVISED REGULARIZER

To guarantee disentanglement, we incorporate supervision when training the BGM, following the similar idea in Locatello et al. (2020b) but with a different formulation. Specifically, let $\xi \in \mathbb{R}^m$ be the underlying ground-truth factors of interests of $x$, following distribution $p_\xi$, and $y_i$ be some continuous or discrete observation of the underlying factor $\xi_i$ satisfying $\xi_i = \mathbb{E}(y_i|x)$ for $i = 1, \ldots, m$. For example, in the case of human face images, $y_1$ can be the binary label indicating whether a person is young or not, and $\xi_1 = \mathbb{E}(y_1|x) = \mathbb{P}(y_1 = 1|x)$ is the probability of being young given one image $x$.

Let $\bar{E}(x)$ be the deterministic part of the stochastic transformation $E(x)$, i.e., $\bar{E}(x) = \mathbb{E}(E(x)|x)$, which is used for representation learning. We consider the following objective:

$$L(E, G) = L_{\text{gen}}(E, G) + \lambda L_{\text{sup}}(E), \tag{2}$$

where $L_{\text{sup}} = \sum_{i=1}^m \mathbb{E}_{x,y}[\text{CE}(\bar{E}_i(x), y_i)]$ if $y_i$ is the binary or bounded continuous label of the $i$-th factor $\xi_i$, where $\text{CE}(l, y) = y \log \sigma(l) + (1 - y) \log(1 - \sigma(l))$ is the cross-entropy loss with $\sigma(\cdot)$ being the sigmoid function; $L_{\text{sup}} = \sum_{i=1}^m \mathbb{E}_{x,y}[\bar{E}_i(x) - y_i]^2$ if $y_i$ is the continuous observation of $\xi_i$, and $\lambda > 0$ is the coefficient to balance both terms. We empirically find the choice of $\lambda$ quite insensitive to different tasks and datasets, and hence set $\lambda = 5$ in all experiments. Estimating of $L_{\text{gen}}$ requires the unlabelled dataset $\{x^1, \ldots, x^N\}$ while estimating $L_{\text{sup}}$ requires a labeled dataset $\{(x^j, y^j) : j = 1, \ldots, N_s\}$ where $N_s$ can be much smaller than $N$.

In contrast, Locatello et al. (2020b) propose the regularizer $L_{\text{sup}} = \sum_{i=1}^m \mathbb{E}_{x,z}[\text{CE}(\bar{E}_i(x), z_i)]$ involving only the latent variable $z$ which is a part of the generative model, without distinguishing from the ground-truth factor $\xi$ and its observation $y$. Hence they do not establish any theoretical justification on disentanglement. Besides, they adopt a VAE loss for $L_{\text{gen}}$ with an independent prior, which suffers from the unidentifiability problem described in the next section.

### 3.3 UNIDENTIFIABILITY WITH AN INDEPENDENT PRIOR

Intuitively, the above supervised regularizer aims at ensuring some alignment between factor $\xi$ and latent variable $z$. We start with the definition of a disentangled representation following this intuition.

**Definition 1** (Disentangled representation). *Given the underlying factor $\xi \in \mathbb{R}^m$ of data $x$, a deterministic encoder $E$ is said to learn a disentangled representation with respect to $\xi$ if $\forall i = 1, \ldots, m$, there exists a 1-1 function $g_i$ such that $E_i(x) = g_i(\xi_i)$. Further, a stochastic encoder $E$ is said to be disentangled wrt $\xi$ if its deterministic part $\bar{E}(x)$ is disentangled wrt $\xi$.*

As stated above, we consider the general case where the underlying factors of interests are causally correlated. Then the goal becomes to disentangle the causal factors. Previous methods mostly use an independent prior for $z$, which contradicts with the truth. We make this formal through the following proposition, which indicates that the disentangled representation is generally unidentifiable with an independent prior.

**Proposition 1.** *Let $E^*$ be any encoder that is disentangled wrt $\xi$. Let $b^* = L_{\text{sup}}(E^*)$, $a = \min_G L_{\text{gen}}(E^*, G)$, and $b = \min_{\{(E,G):L_{\text{gen}}=0\}} L_{\text{sup}}(E)$. Assume the elements of $\xi$ are connected by a causal graph whose adjacency matrix $A_0$ is not a zero matrix. Suppose the prior $p_z$ is factorized, i.e., $p_z(z) = \prod_{i=1}^k p_i(z_i)$. Then we have $a > 0$, and either when $b^* \geq b$ or $b^* < b$ and $\lambda < \frac{a}{b - b^*}$, there exists a solution $(E', G')$ such that for any generator $G$, we have $L(E', G') < L(E^*, G)$.*

All proofs are given in Appendix A. This proposition directly suggests that minimizing (2) favors the solution $(E', G')$ over one with a disentangled encoder $E^*$. Thus, with an independent prior we have no way to identify the disentangled solution with $\lambda$ that is not large enough. However, in real applications it is impossible to estimate the threshold, and too large $\lambda$ makes it difficult to learn the BGM. In the following section we propose a solution to this problem.

## 4 CAUSAL DISENTANGLEMENT LEARNING

### 4.1 GENERATIVE MODEL WITH A CAUSAL PRIOR

We propose to use a causal model as the prior $p_z$. Specifically we use the generalized nonlinear Structural Causal Model (SCM) proposed by Yu et al. (2019) as follows

$$z = f((I - A^\top)^{-1}h(\epsilon)) := F_\beta(\epsilon), \tag{3}$$

where $A$ is the weighted adjacency matrix of the directed acyclic graph (DAG) upon the $k$ elements of $z$ (i.e., $A_{ij} \neq 0$ if and only if $z_i$ is the parent of $z_j$), $\epsilon$ denotes the exogenous variables following $\mathcal{N}(0, I)$, $f$ and $h$ are element-wise nonlinear transformations, and $\beta = (f, h, A)$ denotes the set of parameters of $f$, $h$ and $A$, with the parameter space $\mathcal{B}$. Further let $\mathbf{1}_A = \mathbf{I}(A \neq 0)$ denote the corresponding binary adjacency matrix, where $\mathbf{I}$ is the element-wise indicator function.

When $f$ is invertible, (3) is equivalent to

$$f^{-1}(z) = A^\top f^{-1}(z) + h(\epsilon) \tag{4}$$

which indicates that the factors $z$ satisfy a linear SCM after nonlinear transformation $f$, and enables interventions on latent variables as discussed later. The model structure is presented in Figure 1. Note that different from our model where $z$ is the latent variable following the prior (3) with the goal of causal disentanglement, Yu et al. (2019) proposed a causal discovery method where variables $z$ are observed with the aim of learning the causal structure among $z$.

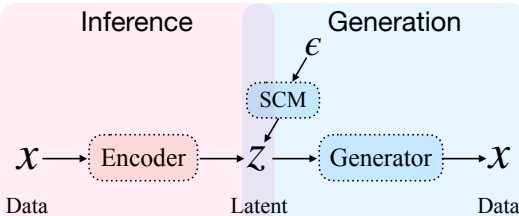

Figure 1: Model structure of a bidirectional generative model (BGM) with an SCM prior.

In causal structure learning, the graph is required to be acyclic. Zheng et al. (2018) propose an equality constraint whose satisfaction ensures acyclicity and solve the problem with augmented Lagrangian method, which however leads to optimization difficulties (Ng et al., 2020). In this paper, to avoid dealing with the non-convex constraint but focus on disentangling, we assume to have some prior knowledge of the binary causal structure. Specifically, we assume the super-graph of the true binary graph $\mathbf{1}_{A^*}$ is given, the best case of which is the true graph while the worst is that only the causal ordering is available. Then we learn the weights of the non-zero elements of the prior adjacency matrix that indicate the sign and scale of causal effects, jointly with other parameters using the formulation and algorithm described in later sections. To incorporate structure learning methods and jointly learn the structure from scratch with guarantee of identifiability could be explored in future work. An ablation study is done in Appendix B regarding this prior knowledge.

To enable causal controllable generation, we use invertible $f$ and $h$ and describe the mechanism to generate images from interventional distributions of latent variables. Note that interventions can be formalized as operations that modify a subset of equations in (4) (Pearl et al., 2000). Suppose we would like to intervene on the $i$-th dimension of $z$, i.e., $\text{Do}(z_i = c)$, where $c$ is a constant. Once we have latent factors $z$ inferred from data $x$, i.e., $z = E(x)$, or sampled from prior $p_z$, we follow the intervened equations in (4) to obtain $z'$ on the left hand side using ancestral sampling by performing (4) iteratively. Then we decode the intervened latent factor $z'$ to generate the sample $G(z')$. In Section 5.1 we define the two types of interventions of most interests in applications.

Another issue of the model is the latent dimension, to handle which we propose the so-called composite prior. Recall that $m$ is the number of generative factors that we are interested to disentangle,

e.g., all the semantic concepts related to some field, where $m$ tends to be smaller than the total number $M$ of generative factors. The latent dimension $k$ of the generative model should be no less than $M$ to allow a sufficient degree of freedom in order to generate or reconstruct data well. Since $M$ is generally unknown in reality, we set a sufficiently large $k$, at least larger than $m$ which is a trivial lower bound of $M$. The role of the remaining $k - m$ dimensions is to capture other factors necessary for generation whose structure is not cared or explicitly modeled. Then we propose to use a prior that is a composition of a causal model for the first $m$ dimensions and another distribution for the other $k - m$ dimensions to capture other factors necessary for generation, like a standard Gaussian.

## 4.2 FORMULATION AND IDENTIFIABILITY OF DISENTANGLEMENT

In this section, we present the formulation of DEAR and establish the theoretical justification of it. Compared with the BGM described in Section 3.1, here we have one more module to learn that is the SCM prior. Thus $p_G(x, z)$ becomes $p_{G,F}(x, z) = p_F(z)p_G(x|z)$ where $p_F(z)$ or $p_\beta(z)$ denotes the marginal distribution of $F_\beta(\epsilon)$ with $\epsilon \sim \mathcal{N}(0, I)$. We then rewrite the generative loss as follows

$$L_{\text{gen}}(E, G, F) = D_{\text{KL}}(q_E(x, z), p_{G,F}(x, z)). \tag{5}$$

Then we propose the following formulation to learn causal generative causal representations:

$$\min_{E,G,F} L(E, G, F) := L_{\text{gen}}(E, G, F) + \lambda L_{\text{sup}}(E). \tag{6}$$

In order to achieve causal disentanglement, we make two assumptions on the causal model. Assumption 1 supposes a sufficiently large capacity of the SCM in (3) to contain the underlying distribution $p_\xi$, which is reasonable due to the generalization of the nonlinear SCM. Assumption 2 states the identifiability of the true causal structure $\mathbf{1}_{A_0}$ of $\xi$, which is applicable given the true causal ordering under basic Markov and causal minimality conditions (Pearl, 2014; Zhang & Spirtes, 2011).

**Assumption 1** (SCM capacity). *The underlying distribution $p_\xi$ belongs to the distribution family $\{p_\beta : \beta \in \mathcal{B}\}$, i.e., there exits $\beta_0 = (f_0, h_0, A_0)$ such that $p_\xi = p_{\beta_0}$.*

**Assumption 2** (Structure identifiability). *For all $\beta = (f, h, A) \in \mathcal{B}$ with $p_\beta = p_{\beta_0}$, it holds that $\mathbf{1}_A = \mathbf{1}_{A_0}$.*

The following theorem then guarantees that under appropriate conditions the DEAR formulation can learn the disentangled representations defined in Definition 1.

**Theorem 1.** *Assume the infinite capacity of $E$ and $G$. Further under Assumption 1-2, DEAR formulation (6) learns the disentangled encoder $E^*$. Specifically, we have $g_i(\xi_i) = \sigma^{-1}(\xi_i)$ if CE loss is used for the supervised regularizer, and $g_i(\xi_i) = \xi_i$ if $L_2$ loss is used.*

Note that the identifiability we establish in this paper differs from some previous work on the parameter identifiability, e.g., Khemakhem et al. (2020). We argue that to learn disentangled representations, the form in Definition 1, *i.e.*, the existence but not the uniqueness of $g_i$'s, is sufficient to identify the relation among the representations and the data. In contrast, parameter identifiability may not be achievable in many cases like over-parametrization. Thus the identifiability discussed here is more realistic in terms of the goal of disentangling. Later we provide empirical evidence to support the theory directly through the application in causal controllable generation.

## 4.3 ALGORITHM

In this section we propose the algorithm to solve the formulation (6). The SCM prior $p_F(z)$ and implicit generated conditional $p_G(x|z)$ make (5) lose an analytic form. Hence we adopt a GAN method to adversarially estimate the gradient of (5). We parametrize $E_\phi(x)$ and $G_\theta(z)$ by neural networks. Different from Shen et al. (2020), the prior also involves learnable parameters. We present in the following lemma the gradient formulas of (5).

**Lemma 1.** *Let $r(x, z) = q(x, z)/p(x, z)$ and $\mathcal{D}(x, z) = \log r(x, z)$. Then we have*

$$\nabla_\theta L_{\text{gen}} = -\mathbb{E}_{z \sim p_\beta(z)}[s(x, z)\nabla_x \mathcal{D}(x, z)^\top|_{x=G_\theta(z)}\nabla_\theta G_\theta(z)],$$

$$\nabla_\phi L_{\text{gen}} = \mathbb{E}_{x \sim q_x}[\nabla_z \mathcal{D}(x, z)^\top|_{z=E_\phi(x)}\nabla_\phi E_\phi(x)], \tag{7}$$

$$\nabla_\beta L_{\text{gen}} = -\mathbb{E}_\epsilon[s(x, z)(\nabla_x \mathcal{D}(x, z)^\top \nabla_\beta G(F_\beta(\epsilon)) + \nabla_z \mathcal{D}(x, z)^\top \nabla_\beta F_\beta(\epsilon))|_{z=F_\beta(\epsilon)}^{x=G(F_\beta(\epsilon))}],$$

*where $s(x, z) = e^{\mathcal{D}(x,z)}$ is the scaling factor.*

We then estimate the gradients in (7) by training a discriminator $D_\psi$ via empirical logistic regression: $\min_{\psi'}[\frac{1}{|S_e|}\sum_{(x,z)\in S_e}\log(1+e^{-D_{\psi'}(x,z)}) + \frac{1}{|S_g|}\sum_{(x,z)\in S_g}\log(1+e^{D_{\psi'}(x,z)})]$, where $S_e$ and $S_g$ are finite samples from $q_E(x,z)$ and $p_G(x,z)$ respectively, leading to a GAN approach.

Based on above, we propose Algorithm 1 to learn disentangled generative causal representation.

---

**Algorithm 1:** Disentangled gEnerative cAusal Representation (DEAR) Learning

---

**Input:** training set $\{x_1, \ldots, x_N, y_1, \ldots, y_{N_s}\}$, initial parameters $\phi, \theta, \beta, \psi$, batch-size $n$

1  **while** *not convergence* **do**
2     **for** *multiple steps* **do**
3        Sample $\{x_1, \ldots, x_n\}$ from the training set, $\{\epsilon_1, \ldots, \epsilon_n\}$ from $\mathcal{N}(0, I)$
4        Generate from the causal prior $z_i = F_\beta(\epsilon_i), i = 1, \ldots n$
5        Update $\psi$ by descending the stochastic gradient:

        $\frac{1}{n}\sum_{i=1}^n \nabla_\psi \left[\log(1+e^{-D_\psi(x_i, E_\phi(x_i))}) + \log(1+e^{D_\psi(G_\theta(z_i), z_i)})\right]$

6     Sample $\{x_1, \ldots, x_n, y_1, \ldots, y_{n_s}\}$, $\{\epsilon_1, \ldots, \epsilon_n\}$ as above; generate $z_i = F_\beta(\epsilon_i)$
7     Compute $\theta$-gradient: $-\frac{1}{n}\sum_{i=1}^n s(G_\theta(z_i), z_i)\nabla_\theta D_\psi(G_\theta(z_i), z_i)$
8     Compute $\phi$-gradient: $\frac{1}{n}\sum_{i=1}^n \nabla_\phi D_\psi(x_i, E_\phi(x_i)) + \frac{1}{n_s}\sum_{i=1}^{n_s}\nabla_\phi L_{\sup}(\phi; x_i, y_i)$
9     Compute $\beta$-gradient: $-\frac{1}{n}\sum_{i=1}^n s(G(z_i), z_i)\nabla_\beta D_\psi(G_\theta(F_\beta(\epsilon_i)), F_\beta(\epsilon_i))$
10    Update parameters $\phi, \theta, \beta$ using the gradients

**Return:** $\phi, \theta, \beta$

---

Remark: without loss of generality, assume the first $N_s$ samples in the training set and the first $n_s$ samples in each mini-batch has available labels; $n_s$ may vary across different iterations.

## 5 EXPERIMENTS

We evaluate our methods on two datasets. The first one is a synthesized dataset Pendulum similar to the one in Yang et al. (2020). As shown in Figure 3, each image is generated by four continuous factors: *pendulum_angle*, *light_angle*, *shadow_length* and *shadow_position* whose underlying structure is given in Figure 2(a) following physical mechanisms. To make the dataset realistic, we introduce random noises when generating the two effects from the causes, representing the measurement error. We further introduce 20% corrupted data whose shadow is randomly generated, mimicking some environmental disturbance. The sample sizes for training, validation and test set are all 6,724.[1]

The second one is a real human face dataset CelebA (Liu et al., 2015), containing 202,599 images with 40 labelled binary attributes. Among them we consider two groups of causally correlated factors shown in 2(b,c). We believe these two datasets are diverse enough to assess our methods. All the details of experimental setup and architectures are given in Appendix C.

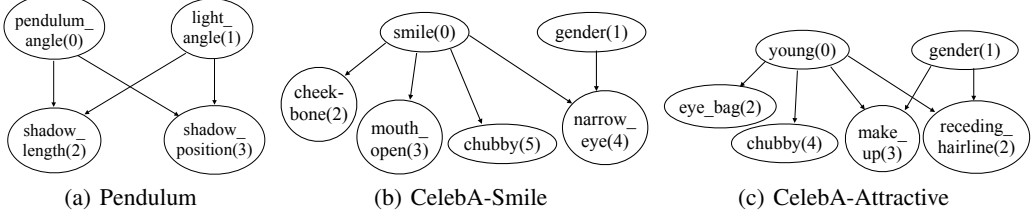

(a) Pendulum         (b) CelebA-Smile         (c) CelebA-Attractive

Figure 2: Underlying causal structures.

### 5.1 CONTROLLABLE GENERATION

We first investigate the performance of our methods in disentanglement through applications in causal controllable generation (CG). Traditional CG methods mainly manipulate the independent generative factors (Karras et al., 2019), while we consider the general case where the factors are causally correlated. With a learned SCM as the prior, we are able to generate images from any desired interventional distributions of the latent factors. For example, we can manipulate only the

---

[1]The Pendulum dataset will be released as a causal disentanglement benchmark soon.

cause factor while leave its effects unchanged. Besides, the BGM framework enables controllable generation either from scratch or a given unlabeled image.

We consider two types of intervention. In traditional traversals, we manipulate one dimension of the latent vector while keep the others fixed to either their inferred or sampled values (Higgins et al., 2017). A causal view of such operations is an intervention on all the variables by setting them as constants with only one of them varying. Another interesting type of interventional distribution is to intervene on only one latent variable, *i.e.*, $\mathbb{P}_{\mathrm{do}(z_i=c)}(z)$. The proposed SCM prior enables us to conduct such intervention though the mechanism given in Section 4.1.

Figure 3-4 illustrate the results of causal controllable generation of the proposed DEAR and the baseline method with an independent prior, S-$\beta$-VAE (Locatello et al., 2020b). Results from other baselines including S-TCVAE, S-FactorVAE (which essentially make no difference due to the independence assumption) and CausalGAN are given in Appendix D. Note that we do not compare with unsupervised disentanglement methods because of fairness and their lack of justification.

In each figure, we first infer the latent representations from a test image in block (c). The traditional traversals of two models are given in blocks (a,b). We see that in each line when manipulating one latent dimension, the generated images from our model vary only in a single factor, indicating that our method can disentangle the causally correlated factors. It is worth pointing out that we are the first to achieve the disentanglement between the cause and its effects, while other methods tend to entangle them. In block (d), we show the results of intervention on the latent variables representing the cause factors, which clearly show that intervening on a cause variable changes its effect variables. Results in Appendix D further show that intervening on an effect node does not influence its cause.

Since the underlying factors are causally correlated, all previous quantitative metrics for disentanglement no longer apply. We provide more qualitative traversals in Appendix D to show the overall performance. A quantitative metric for causal disentanglement is worth exploring in future work.

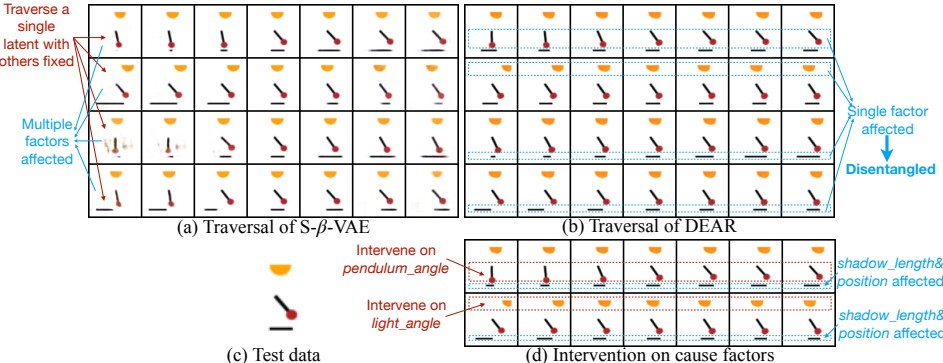

Figure 3: Results of causal controllable generation on Pendulum.

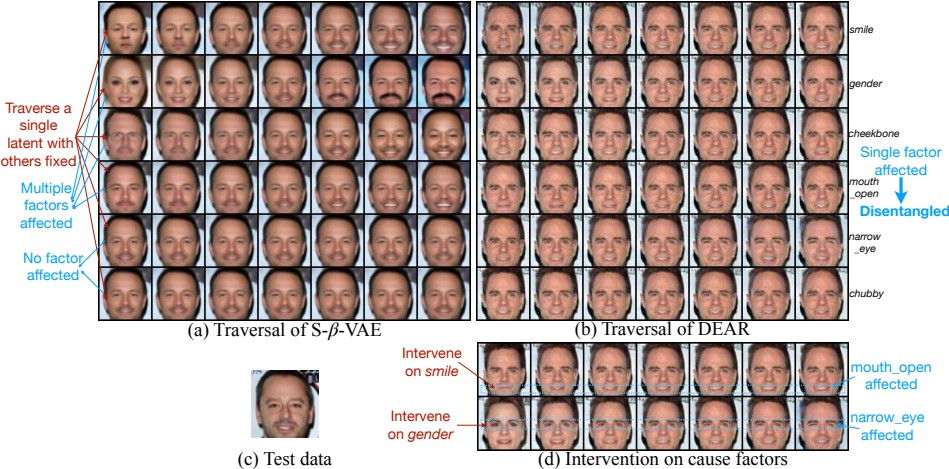

Figure 4: Results of causal controllable generation on CelebA.

Table 1: Sample efficiency and test accuracy with different training sample sizes. DEAR-lin and -nlr denote the model with linear and nonlinear $f$. Line 1 is unsupervised; 2-3 are semi-supervised; others are supervised.

| | (a) CelebA | | | (b) Pendulum | | |
|---|---|---|---|---|---|---|
| **Method** | **100(%)** | **10,000(%)** | **Eff(%)** | **100(%)** | **all(%)** | **Eff(%)** |
| ResNet | $68.06_{\pm0.19}$ | $79.51_{\pm0.31}$ | $85.59_{\pm0.27}$ | $79.71_{\pm0.98}$ | $90.64_{\pm1.57}$ | $87.97_{\pm2.11}$ |
| DEAR-lin-10% | $78.09_{\pm0.59}$ | $79.54_{\pm0.41}$ | $98.18_{\pm0.49}$ | $88.93_{\pm1.40}$ | $93.18_{\pm0.18}$ | $95.43_{\pm1.33}$ |
| DEAR-nlr-10% | $80.30_{\pm0.24}$ | $80.87_{\pm0.12}$ | $\mathbf{99.29}_{\pm0.23}$ | $87.65_{\pm0.46}$ | $91.27_{\pm0.21}$ | $96.03_{\pm0.29}$ |
| ResNet-pretrain | $76.84_{\pm2.08}$ | $83.75_{\pm0.93}$ | $91.74_{\pm1.98}$ | $79.59_{\pm0.93}$ | $89.16_{\pm1.60}$ | $89.28_{\pm0.59}$ |
| S-VAE | $77.07_{\pm1.42}$ | $79.87_{\pm1.67}$ | $96.49_{\pm1.68}$ | $84.16_{\pm0.69}$ | $90.89_{\pm0.28}$ | $92.60_{\pm0.49}$ |
| S-$\beta$-VAE | $71.78_{\pm1.99}$ | $76.63_{\pm0.24}$ | $93.67_{\pm2.41}$ | $79.95_{\pm1.65}$ | $87.87_{\pm0.52}$ | $90.98_{\pm1.47}$ |
| S-TC-VAE | $77.10_{\pm2.08}$ | $81.63_{\pm0.20}$ | $94.45_{\pm2.72}$ | $85.36_{\pm1.11}$ | $90.33_{\pm0.33}$ | $94.51_{\pm1.31}$ |
| DEAR-lin | $83.51_{\pm0.77}$ | $84.92_{\pm0.11}$ | $98.34_{\pm0.81}$ | $90.21_{\pm0.94}$ | $\mathbf{93.31}_{\pm0.14}$ | $96.68_{\pm0.89}$ |
| DEAR-nlr | $\mathbf{84.44}_{\pm0.48}$ | $\mathbf{85.10}_{\pm0.09}$ | $99.23_{\pm0.51}$ | $\mathbf{90.62}_{\pm0.32}$ | $92.57_{\pm0.08}$ | $\mathbf{97.93}_{\pm0.29}$ |

## 5.2 DOWNSTREAM TASK

The previous section verifies the good disentanglement performance of DEAR. In this section, equipped with DEAR, we investigate and demonstrate the benefits of learned disentangled causal representations in sample efficiency and distributional robustness.

We state the downstream tasks. On CelebA, we consider the structure CelebA-Attractive in Figure 2(c). We artificially create a target label $\tau = 1$ if *young*=1, *gender*=0, *receding_hairline*=0, *make_up*=1, *chubby*=0, *eye_bag*=0, and $\tau = 0$ otherwise, indicating one kind of attractiveness as a slim young woman with makeup and thick hair.[2] On the pendulum dataset, we regard the label of data corruption as the target $\tau$, *i.e.*, $\tau = 1$ if the data is corrupted and $\tau = 0$ otherwise. We consider the downstream tasks of predicting the target label. In both cases, the factors of interests in Figure 2(a,c) are causally related to $\tau$, which are the features that humans use to do the task. Hence it is conjectured that a disentangled representation of these causal factors tends to be more data efficient and invariant to distribution shifts.

### 5.2.1 SAMPLE EFFICIENCY

For a BGM including the previous state-of-the-art supervised disentangling methods S-VAEs (Locatello et al., 2020b) and DEAR, we use the learned encoder to embed the training data to the latent space and train a MLP classifier on the representations to predict the target label. Without an encoder, one normally needs to train a convolutional neural network with raw images as the input. Here we adopt the ResNet50 as the baseline classifier which is the architecture of the BGM encoder. Since disentangling methods use additional supervision of the generative factors, we consider another baseline that is pretrained using multi-label prediction of the factors on the same training set.

To measure the sample efficiency, we use the statistical efficiency score defined as the average test accuracy based on 100 samples divided by the average accuracy based on 10,000/all samples, following Locatello et al. (2019). Table 1 presents the results, showing that DEAR owns the highest sample efficiency on both datasets. ResNet with raw data inputs has the lowest efficiency, although multi-label pretraining improves its performance to a limited extent. S-VAEs have better efficiency than the ResNet baselines but lower accuracy under the case with more training data, which we think is mainly because the independent prior conflicts with the supervised loss as indicated in Proposition 1, making the learned representations entangled (as shown in the previous section) and less informative. Besides, we also investigate the performance of DEAR under the semi-supervised setting where only 10% of the labels are available. We find that DEAR with fewer labels has comparable sample efficiency with that in the fully supervised setting, with a sacrifice in accuracy that is yet still comparable to other baselines with more supervision.

---

[2]Note that the definition of attractiveness here only refers to one kind of attractiveness, which has nothing to do with the linguistic definition of attractiveness.

Table 2: Distributional robustness. The worst-case and average test accuracy

| | (a) CelebA | | (b) Pendulum | |
|---|---|---|---|---|
| **Method** | **WorstAcc(%)** | **AvgAcc(%)** | **WorstAcc(%)** | **AvgAcc(%)** |
| ResNet | $59.12_{\pm 1.78}$ | $82.12_{\pm 0.26}$ | $60.48_{\pm 2.73}$ | $87.40_{\pm 0.89}$ |
| DEAR-lin-10% | $71.40_{\pm 0.47}$ | $81.04_{\pm 0.14}$ | $63.93_{\pm 1.33}$ | $89.70_{\pm 0.63}$ |
| DEAR-nlr-10% | $70.44_{\pm 1.02}$ | $81.94_{\pm 0.31}$ | $65.59_{\pm 1.90}$ | $90.19_{\pm 0.63}$ |
| ResNet-multi | $59.17_{\pm 4.02}$ | $82.05_{\pm 0.25}$ | $61.70_{\pm 4.02}$ | $87.20_{\pm 1.00}$ |
| S-VAE | $60.54_{\pm 3.48}$ | $79.51_{\pm 0.58}$ | $20.78_{\pm 4.45}$ | $84.26_{\pm 1.31}$ |
| S-$\beta$-VAE | $63.85_{\pm 2.09}$ | $80.82_{\pm 0.19}$ | $44.12_{\pm 9.73}$ | $86.99_{\pm 1.78}$ |
| S-TC-VAE | $64.93_{\pm 3.30}$ | $81.58_{\pm 0.14}$ | $35.50_{\pm 5.57}$ | $86.64_{\pm 1.15}$ |
| DEAR-lin | $\mathbf{76.05}_{\pm 0.70}$ | $83.56_{\pm 0.09}$ | $\mathbf{74.95}_{\pm 1.26}$ | $\mathbf{93.61}_{\pm 0.13}$ |
| DEAR-nlr | $71.37_{\pm 0.66}$ | $\mathbf{83.81}_{\pm 0.08}$ | $72.48_{\pm 0.74}$ | $93.11_{\pm 0.14}$ |

### 5.2.2 DISTRIBUTIONAL ROBUSTNESS

We manipulate the training data to inject spurious correlations between the target label and some spurious attributes. On CelebA, we regard *mouth_open* as the spurious factor; on Pendulum, we choose *background_color* $\in$ {blue(+), white(−)}. We manipulate the training data such that the target label is more strongly correlated with the spurious attributes, *i.e.*, the target label and the spurious attribute of 80% of the examples are both positive or negative, while those of 20% examples are opposite. For example, in the manipulated training set, 80% smiling examples in CelebA have an open mouth; 80% corrupted examples in Pendulum are masked with a blue background. The test set however does not have these correlations, leading to a distribution shift.

Intuitively these spurious attributes are not causally correlated to the target label, but normal independent and identically distributed (IID) based methods like empirical risk minimization (ERM) tend to exploit these easily learned spurious correlations in prediction, and hence face performance degradation when the such correlation no longer exists during test. In contrast, causal factors are regarded invariant and thus robust under such shifts. Previous sections justify both theoretically and empirically that DEAR can learn disentangled causal representations. We then apply those representations by training a classifier upon them, which is conjectured to be invariant and robust. Baseline methods include ERM, multi-label ERM to predict target label and all the factors considered in disentangling to have the same amount of supervision, and S-VAEs that can not disentangle well in the causal case.

Table 2 shows the average and worst-case (Sagawa et al., 2019) test accuracy to assess both the overall classification performance and distributional robustness, where we group the test set according to the two binary labels, the target one and the spurious attribute, into four cases and regard the one with the worst accuracy as the worst-case, which usually owns the opposite correlation to the training data. We see that the classifiers trained upon DEAR representations outperform the baselines in both metrics. Particularly, when comparing the worst-case accuracy with the average one, we observe a slump from around 80 to around 60 for other methods on CelebA, while DEAR enjoys an acceptable small decline. These results support the above conjecture and the benefits of causal disentanglement in distributional robustness.

## 6 CONCLUSION

This paper showed that previous methods with the independent latent prior assumption fail to learn disentangled representation when the underlying factors of interests are causally correlated. We then proposed a new disentangled learning method called DEAR with theoretical guarantees. Extensive experiments demonstrated the effectiveness of DEAR in causal generation, and the benefits of the learned representations for downstream tasks.

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

## APPENDIX A   PROOFS

### A.1   PROOF OF PROPOSITION 1

*Proof.* On one hand, by the assumption that the elements of $\xi$ are connected by a causal graph whose adjacency matrix is not a zero matrix. Then exist $i \neq j$ such that $\xi_i$ and $\xi_j$ are not independent, indicating that the probability density of $\xi$ cannot be factorized. Since $E^*$ is disentangled wrt $\xi$, by Definition 1, $\forall i = 1, \ldots, m$ there exists $g_i$ such that $E_i^*(x) = g_i(\xi_i)$. This implies that the probability density of $E^*(x)$ is not factorized.

On the other hand, notice that the distribution family of the latent prior is $\{p_z : p_z \text{ is factorized}\}$. Hence the intersection of the marginal distribution families of $z$ and $E^*(x)$ is an empty set. Then the joint distribution families of $(x, E^*(x))$ and $(G(z), z)$ also have an empty intersection.

We know that $L_{\text{gen}}(E, G) = 0$ implies $q_E(x, z) = p_G(x, z)$ which contradicts the above. Therefore, we have $a = \min_G L_{\text{gen}}(E^*, G) > 0$.

Let $(E', G')$ be the solution of the optimization problem $\min_{\{(E,G):L_{\text{gen}}=0\}} L_{\text{sup}}(E)$. Then we have $L' = L(E', G') = \lambda b$, and $L^* = L(E^*, G) \geq a + \lambda b^* > \lambda b^*$ for any generator $G$. When $b^* \geq b$ we directly have $L' < L^*$. When $b^* < b$ and $\lambda$ is not large enough, *i.e.*, $\lambda < \frac{a}{b - b^*}$, we have $L' < L^*$. □

### A.2   PROOF OF THEOREM 1

*Proof.* Assume $E$ is deterministic.

On one hand, for each $i = 1, \ldots, m$, first consider the cross-entropy loss

$$L_{\text{sup},i}(E) = \mathbb{E}_{(x,y)}[\text{CE}(E_i(x), y_i)] = \int p(x)p(y_i|x)(y_i \log \sigma(E_i(x)) + (1 - y_i) \log(1 - \sigma(E_i(x)))) dx dy_i,$$

where $p(y_i|x)$ is the probability mass function of the binary label $y_i$ given $x$, characterized by $\mathbb{P}(y_i = 1|x) = \mathbb{E}(y_i|x)$ and $\mathbb{P}(y_i = 0|x) = 1 - \mathbb{E}(y_i|x)$. Let

$$\frac{\partial L_{\text{sup},i}}{\partial \sigma(E_i(x))} = \int p(x)p(y_i|x) \left( y_i \frac{1}{\sigma(E_i)(1 - \sigma(E_i))} - \frac{1}{1 - \sigma(E_i)} \right) dx dy_i = 0.$$

Then we know that $E_i^*(x) = \sigma^{-1}(\mathbb{E}(y_i|x)) = \sigma^{-1}(\xi_i)$ minimizes $L_{\text{sup},i}$.

Consider the $L_2$ loss

$$L_{\text{sup},i}(\phi) = \mathbb{E}_{(x,y)}[\bar{E}_i(x) - y_i]^2 = \int p(x)p(y_i|x)\|E_i(x) - y_i\|^2 dx dy_i.$$

Let

$$\frac{\partial L_{\text{sup},i}}{\partial E_i(x)} = 2 \int p(x)p(y_i|x)(E_i(x) - y_i) dx dy_i = 0.$$

Then we know that $E_i^*(x) = \mathbb{E}(y_i|x) = \xi_i$ minimizes $L_{\text{sup},i}$ in this case.

On the other hand, by Assumption 1 there exists $\beta_0 = (f_0, h_0, A_0)$ such that $p_\xi = p_{\beta_0}$. Then the distribution of $E^*(x)$ is given by $p_{\beta^*}$ with $\beta^* = (g \circ f_0, h_0, A_0)$. Assumption 2 ensures that there is no $\beta' = (f', h', A')$ such that $A' \neq A_0$ but $p_{\beta'} = p_{\beta^*}$. Let $F^* = F_{\beta^*}$. Further due to the infinite capacity of $G$, we have the distribution family of $p_{G,F^*}(x, z)$ contains $q_{E^*}(x, z)$. Then by minimizing the loss in (6) over $G$, we can find $G^*$ such that $p_{G^*,F^*}(x, z)$ matches $q_{E^*}(x, z)$ and thus $L_{\text{gen}}(E^*, G^*, F^*)$ reaches 0.

Hence minimizing $L = L_{\text{gen}} + \lambda L_{\text{sup}}$, which is the DEAR formulation (6), leads to the solution with $E_i^*(x) = g_i(\xi_i)$ with $g_i(\xi_i) = \sigma^{-1}(\xi_i)$ if CE loss is used, and $g_i(\xi_i) = \xi_i$ if $L_2$ loss is used, and the true binary adjacency matrix.

For a stochastic encoder, we establish the disentanglement of its deterministic part as above, and follow Definition 1 to obtain the desired result. □

### A.3 PROOF OF LEMMA 1

We follow the same proof scheme as in Shen et al. (2020) where the only difference lies in the gradient wrt the prior parameter $\beta$. To make this paper self-contained, we restate some proof steps here using our notations.

Let $\| \cdot \|$ denote the vector 2-norm. For a scalar function $h(x, y)$, let $\nabla_x h(x, y)$ denote its gradient with respect to $x$. For a vector function $g(x, y)$, let $\nabla_x g(x, y)$ denote its Jacobi matrix with respect to $x$. Given a differentiable vector function $g(x) : \mathbb{R}^k \to \mathbb{R}^k$, we use $\nabla \cdot g(x)$ to denote its divergence, defined as

$$\nabla \cdot g(x) := \sum_{j=1}^{k} \frac{\partial [g(x)]_j}{\partial [x]_j},$$

where $[x]_j$ denotes the $j$-th component of $x$. We know that

$$\int \nabla \cdot g(x) dx = 0$$

for all vector function $g(x)$ such that $g(\infty) = 0$. Given a matrix function $w(x) = (w_1(x), \ldots, w_l(x)) : \mathbb{R}^k \to \mathbb{R}^{k \times l}$ where each $w_i(x), i = 1 \ldots, l$ is a $k$-dimensional differentiable vector function, its divergence is defined as $\nabla \cdot w(x) = (\nabla \cdot w_1(x), \ldots, \nabla \cdot w_l(x))$.

To prove Lemma 1, we need the following lemma which specifies the dynamics of the generator joint distribution $p_g(x, z)$ and the encoder joint distribution $p_e(x, z)$, denoted by $p_\theta(x, z)$ and $p_\phi(x, z)$ here.

**Lemma 2.** *Using the definitions and notations in Lemma 1, we have*

$$\nabla_\theta p_{\theta,\beta}(x, z) = -\nabla_x p_{\theta,\beta}(x, z)^\top g_\theta(x) - p_{\theta,\beta}(x, z) \nabla \cdot g_\theta(x), \tag{8}$$

$$\nabla_\phi q_\phi(x, z) = -\nabla_z q_\phi(x, z)^\top e_\phi(z) - q_\phi(x, z) \nabla \cdot e_\phi(z), \tag{9}$$

$$\nabla_\beta p_{\theta,\beta}(x, z) = \nabla_x p_{\theta,\beta}(x, z)^\top \tilde{f}_\beta(x) - \nabla_z p_{\theta,\beta}(x, z)^\top f_\beta(z) - p_{\theta,\beta}(x, z) \nabla \cdot \begin{pmatrix} \tilde{f}_\beta(x) \\ f_\beta(z) \end{pmatrix}, \tag{10}$$

*for all data $x$ and latent variable $z$, where $g_\theta(G_\theta(z, \epsilon)) = \nabla_\theta G_\theta(z, \epsilon)$, $e_\phi(E_\phi(x, \epsilon)) = \nabla_\phi E_\phi(x, \epsilon)$, $f_\beta(F_\beta(\epsilon)) = \nabla_\beta F_\beta(\epsilon)$, and $\tilde{f}_\beta(G(F_\beta(\epsilon))) = \nabla_\beta G(F_\beta(\epsilon))$.*

*Proof of Lemma 2.* We only prove (10) which is the distinct part from Shen et al. (2020).

Let $l$ be the dimension of parameter $\beta$. To simplify notation, let random vector $Z = F_\beta(\epsilon)$ and $X = G(Z) \in \mathbb{R}^d$ and $Y = (X, Z) \in \mathbb{R}^{d+k}$, and let $p$ be the probability density of $Y$. For each $i = 1, \ldots, l$, let $\Delta = \delta e_i$ where $e_i$ is a $l$-dimensional unit vector whose $i$-th component is one and all the others are zero, and $\delta$ is a small scalar. Let $Z' = F_{\beta+\delta}(\epsilon)$, $X' = G(Z')$ and $Y' = (X', Z')$ so that $Y'$ is a random variable transformed from $Y$ by

$$Y' = Y + \begin{pmatrix} \tilde{f}_\beta(X) \\ f_\beta(Z) \end{pmatrix} \Delta + o(\delta).$$

Let $p'$ be the probability density of $Y'$. For an arbitrary $y' = (x', z') \in \mathbb{R}^{d+k}$, let $y' = y + \begin{pmatrix} \tilde{f}_\beta(x) \\ f_\beta(z) \end{pmatrix} \Delta + o(\delta)$ and $y = (x, z)$. Then we have

$$
\begin{aligned}
p'(y') &= p(y) |\det(dy'/dy)|^{-1} \\
&= p(y) |\det(I_d + (\nabla \tilde{f}_\beta(x), \nabla f_\beta(z))^\top \Delta + o(\delta))|^{-1} \\
&= p(y) (1 + \Delta^\top \nabla \cdot (\tilde{f}_\beta(x), f_\beta(z))^\top + o(\delta))^{-1} \\
&= p(y) (1 - \Delta^\top \nabla \cdot (\tilde{f}_\beta(x), f_\beta(z))^\top + o(\delta)) \\
&= p(y) - \Delta^\top p(y') \nabla \cdot (\tilde{f}_\beta(x'), f_\beta(z'))^\top + o(\delta) \\
&= p(y') - \Delta^\top (\tilde{f}_\beta(x'), f_\beta(z')) \cdot \nabla_{x'} p(x', z) - \Delta^\top p(y') (\nabla \cdot \tilde{f}_\beta(x'), \nabla \cdot f_\beta(z'))^\top + o(\delta).
\end{aligned}
$$

Since $y'$ is arbitrary, above implies that

$$p'(x, z) = p(x, z) - \Delta^\top (\tilde{f}_\beta(x), f_\beta(z)) \cdot (\nabla_x p(x, z), \nabla_z p(x, z))^\top \cdot \nabla_x p(x, z)$$
$$- \Delta^\top p(x, z)(\nabla \cdot \tilde{f}_\beta(x'), \nabla \cdot f_\beta(z'))^\top + o(\delta)$$

for all $x \in \mathbb{R}^d, z \in \mathbb{R}^k$ and $i = 1, \ldots, l$, leading to (10) by taking $\delta \to 0$, and noting that $p = p_\beta$ and $p' = p_{\beta+\Delta}$. Similarly we can obtain (8) and (9). □

*Proof of Lemma 1.* Recall the objective $D_{\text{KL}}(q, p) = \int q(x, z) \log(p(x, z)/q(x, z)) dx dz$. Denote its integrand by $\ell(q, p)$. Let $\ell_2'(q, p) = \partial \ell(q, p)/\partial p$. We have

$$\nabla_\beta \ell(q(x, z), p(x, z)) = \ell_2'(q(x, z), p(x, z)) \nabla_\beta p_{\theta,\beta}(x, z)$$

where $\nabla_\beta p_{\theta,\beta}(x, z)$ is computed in Lemma 2.

Besides, we have

$$\nabla_x \cdot [\ell_2'(q, p)p(x, z)\tilde{f}_\beta(x)] = \ell_2'(q, p)p(x, z)\nabla \cdot \tilde{f}_\beta(x)$$
$$+ \ell_2'(q, p)\nabla_x p(x, z) \cdot \tilde{f}_\beta(x)$$
$$+ \nabla_x \ell_2'(q, p)p(x, z)\tilde{f}_\beta(x),$$
$$\nabla_z \cdot [\ell_2'(q, p)p(x, z)f_\beta(z)] = \ell_2'(q, p)p(x, z)\nabla \cdot f_\beta(z)$$
$$+ \ell_2'(q, p)\nabla p(x, z) \cdot f_\beta(z)$$
$$+ \nabla \ell_2'(q, p)p(x, z)f_\beta(z).$$

Thus,

$$\nabla_\beta L_{\text{gen}} = \int \nabla_\beta \ell(q(x, z), p(x, z)) dx dz = \int p(x, z)[\nabla_x \ell_2'(q, p)\tilde{f}_\beta(x) + \nabla_z \ell_2'(q, p)f_\beta(z)]$$

where we can compute $\nabla_x \ell_2'(q, p) = s(x, z)\nabla_x \mathcal{D}(x, z)$ and $\nabla_x \ell_2'(q, p) = s(x, z)\nabla_z \mathcal{D}(x, z)$.

Hence

$$\nabla_\beta L_{\text{gen}} = -\mathbb{E}_{(x,z) \sim p(x,z)} \left[ s(x, z)(\nabla_x \mathcal{D}(x, z)^\top \tilde{f}_\beta(x) + \nabla_z \mathcal{D}(x, z)^\top f_\beta(z)) \right]$$
$$= -\mathbb{E}_\epsilon \left[ s(x, z)(\nabla_x \mathcal{D}(x, z)^\top \nabla_\beta G(F_\beta(\epsilon)) + \nabla_z \mathcal{D}(x, z)^\top \nabla_\beta F_\beta(\epsilon))|_{z=F_\beta(\epsilon)}^{x=G(F_\beta(\epsilon))} \right].$$

where the second equality follows reparametrization. □

**Lemma 3.** *For any $a, b \in \mathbb{R}$ ($a < b$), the set of continuous piece-wise linear function $P$ is dense in $\mathcal{C}[a, b]$ where the metric $d(f, g) = \sup_{x \in [a,b]} |f(x) - g(x)|$. Note that $P$ is defined as*

$$P = \cup_{h \in \{(b-a)/n | n \in \mathbb{N}^+\}} P_h$$

$$P_h = \left\{ k + \sum_{i=0}^{(b-a)/h-1} w_i(x - a - ih)\mathbf{1}(x \geq a + ih) \middle| w_i, k \in \mathbb{R} \right\},$$

*where $[\cdot]$ here is floor function.*

*Proof.* Since $[a, b]$ is compact, any function $f \in \mathcal{C}[a, b]$ is uniform continuous, *i.e.*, $\forall \epsilon > 0$, there exists $\delta > 0$ such that

$$|x - y| < \delta \implies |f(x) - f(y)| < \epsilon/2.$$

Let $[a, b] = \cup_{n=0}^{N-1}[a_n, b_n]$, and $g_n(x)$ be a linear function, such that

$$a_n = a + nh,$$
$$b_n = a + (n+1)h,$$
$$g_n(a_n) = f(a_n),$$
$$g_i(b_n) = f(b_n),$$
$$Nh = b - a.$$

Assume that $h < \delta$. For any $x \in [a_n, b_n]$, we have

$$
\begin{aligned}
|f(x) - g_i(x)| &\leq \min\{|f(x) - f(a_n)| + |g_i(x) - g_i(a_n)|, |f(x) - f(b_n)| + |g_i(x) - g_i(b_n)|\} \\
&\leq |g_i(a_n) - g_i(b_n)| + \min\{|f(x) - f(a_n)|, |f(x) - f(b_n)|\} \\
&\leq |f(a_n) - f(b_n)| + \min\{|f(x) - f(a_n)|, |f(x) - f(b_n)|\} \\
&< \epsilon.
\end{aligned}
$$

Thus,

$$
\sup_{x \in [a_n, b_n]} |f(x) - g_n(x)| < \epsilon.
$$

We define

$$
g(x) = \sum_{n=1}^{N-1} g_n(x) \mathbf{1}(x \in [a_n, b_n])
$$

which is obvious that $g(x) \in P_h \subset P$. And we have

$$
\sup_{x \in [a, b]} |f(x) - g(x)| < \epsilon
$$

Therefore, $P$ is dense in $\mathcal{C}[a, b]$ and $P_h$ is $\epsilon$-dense.

$\square$

## APPENDIX B  LEARNING THE STRUCTURE

As mentioned in Section 4.1, our DEAR algorithm requires the prior knowledge on the super-graph of the true graph over the underlying factors of interests. The experiments shown in the main text are all based on the assumption that the true graph is given. In this section we investigate the performance of the learned weighted adjacency matrix and present an ablation study on different extents of prior knowledge on the structure.

### B.1  GIVEN THE TRUE GRAPH

Figure 5 shows the learned weighted adjacency matrices when the true binary structure is given, whose weights show sensible signs and scalings consistent with common knowledge. For example, *smile* and its effect *mouth_open* are positively correlated. The corresponding element in the weighted adjacency $A_{03}$ of (a) turns out positive, which makes sense. Also *gender* (the logit of male) and its effect *make_up* are negatively correlated. Then $A_{13}$ of (b) turns out negative.

### B.2  GIVEN THE TRUE CAUSAL ORDERING

Consider the Pendulum dataset, whose ground-truth structure is given in Figure 2(a). Consider a causal ordering *pendulum_angle*, *light_angle*, *shadow_position*, *shadow_length*, given which we start with a full graph whose elements are randomly initialized around 0 as shown in Figure 6(a). Figure 6 presents the adjacency matrices learned by DEAR at different training epochs, from which we see that it eventually obtains the learned structure that nearly coincides with the one learned given the true graph shown in Figure 5(c). This experiment shows the potential of DEAR to incorporate structure learning methods to learn the latent causal structure from scratch, which will be explored in future research.

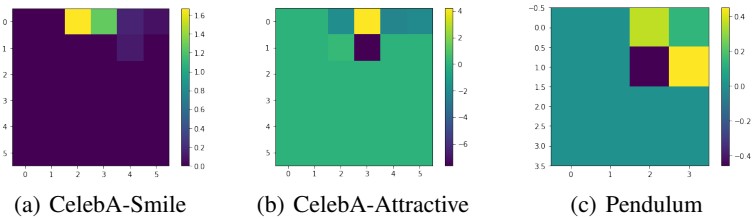

| (a) CelebA-Smile | (b) CelebA-Attractive | (c) Pendulum |

Figure 5: Learned adjacency matrices for different underlying structures.

## APPENDIX C    IMPLEMENTATION DETAILS

In this section we state the details of experimental setup and the network architectures used for all experiments.

**Preprocessing and hyperparameters.**    We pre-process the images by taking a center crops of $128 \times 128$ for CelebA and resizing all images in CelebA and Pendulum to the $64 \times 64$ resolution. We adopt Adam with $\beta_1 = 0$, $\beta_2 = 0.999$, and a learning rate of $1 \times 10^{-4}$ for $D$, $5 \times 10^{-5}$ for $E$, $G$ and $F$, and $1 \times 10^{-3}$ for the adjacency matrix $A$. We use a mini-batch size of 128. For adversarial training in Algorithm 1, we train the $D$ once on each mini-batch. The coefficient $\lambda$ of the supervised regularizer is set to 5. We use CE supervised loss for both CelebA with binary observations of the underlying factors and Pendulum with bounded continuous observations. Note that $L_2$ loss works comparable to CE loss on Pendulum. In downstream tasks, for BGMs with an encoder, we train a two-level MLP classifier with 100 hidden nodes using Adam with a learning rate of $1 \times 10^{-2}$ and a mini-batch size of 128. Models were trained for around 150 epochs on CelebA and 600 epochs on Pendulum on NVIDIA RTX 2080 Ti.

**Network architectures.**    We follow the architectures used in Shen et al. (2020). Specifically, for such realistic data, we adopt the SAGAN (Zhang et al., 2019) architecture for $D$ and $G$. The $D$ network consists of three modules as shown in Figure 7 and detailed described in (Shen et al., 2020). Details for newtork $G$ and $D_x$ are given in Figure 7 and Table 3. The encoder architecture is the ResNet50 (He et al., 2016) followed by a 4-layer MLP of size 1024.

**Implementation of the SCM.**    Recall the nonlinear SCM as the prior

$$Z = f((I - A^\top)^{-1} h(\epsilon)) := F_\beta(\epsilon).$$

We find Gaussians are expressive enough as unexplained noises, so we set $h$ as the identity mapping. As mentioned in Section 4.1 we require the invertibility of $f$. We implement both linear and nonlinear ones. For a linear $f$, we formally refer to

$$f(z) = Wz + b,$$

where $W$ and $b$ are learnable weights and biases. Note that $W$ is a diagonal matrix to model the element-wise transformation. Its inverse function can be easily computed by

$$f^{-1}(z) = W^{-1}(z - b).$$

For a non-linear $f$, we use piece-wise linear functions defined by

$$f^{(i)}(z^{(i)}) = w_0^{(i)} z^{(i)} + \sum_{t=1}^{N_a} w_t^{(i)}(z^{(i)} - a_i)\mathbf{1}(z^{(i)} \geq a_i) + b^{(i)}$$

where $\cdot^{(i)}$ denote the $i$-th dimension of a vector or a vector-function, $a_0 < a_1 < \cdots < a_{N_a}$ are points of division, and $\mathbf{1}(\cdot)$ is the indicator function. From its denseness shown in lemma 3, the family of such piece-wise linear functions is expressive enough to model general element-wise nonlinear invertible transformations.

**Experimental details for baseline methods.**    We reproduce the S-VAEs including S-VAE, S-$\beta$-VAE and S-TCVAE using $E$ and $G$ with the same architecture as ours and adopt the same optimization algorithm for training. The coefficient for the independence regularizer is set to 4 since we

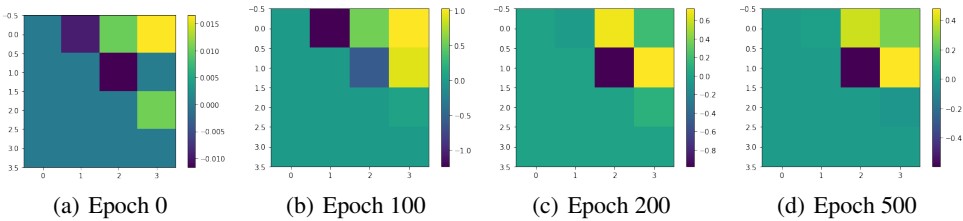

|  (a) Epoch 0  |  (b) Epoch 100  |  (c) Epoch 200  |  (d) Epoch 500  |

Figure 6: Learned adjacency matrices at different training epochs, starting from a random initialization.

notice that setting a larger independence regularizer hurts disentanglement in the correlated case. For the supervised regularizer, we use $\lambda = 1000$ for a balance of generative model and supervision. The ERM ResNet is trained using the same optimizer with a learning rate of $1 \times 10^{-4}$.

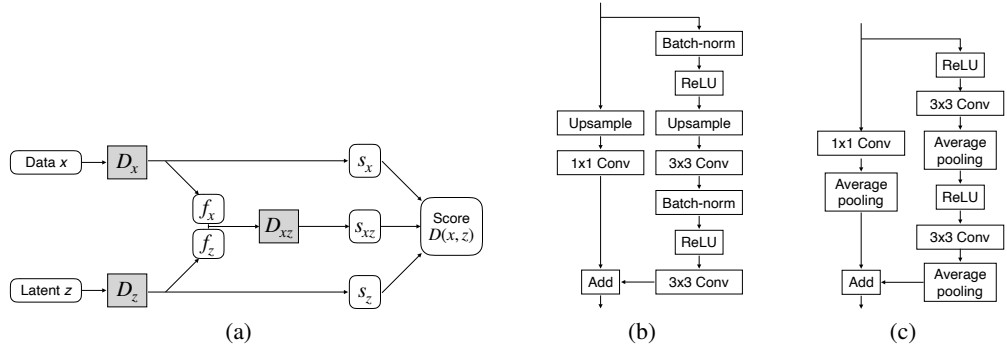

Figure 7: (a) Architecture of the discriminator $D(x, z)$; (b) A residual block (up scale) in the SAGAN generator where we use nearest neighbor interpolation for *Upsampling*; (c) A residual block (down scale) in the SAGAN discriminator.

Table 3: SAGAN architecture ($k = 100$ and $ch = 32$).

| (a) Generator | (b) Discriminator module $D_x$ |
|---|---|
| Input: $z \in \mathbb{R}^k \sim \mathcal{N}(0, I)$ | Input: RGB image $x \in \mathbb{R}^{64 \times 64 \times 3}$ |
| Linear $\to 4 \times 4 \times 16ch$ | ResBlock down $ch \to 2ch$ |
| ResBlock up $16ch \to 16ch$ | Non-Local Block ($64 \times 64$) |
| ResBlock up $16ch \to 8ch$ | ResBlock down $2ch \to 4ch$ |
| ResBlock up $8ch \to 4ch$ | ResBlock down $4ch \to 8ch$ |
| Non-Local Block ($64 \times 64$) | ResBlock down $8ch \to 16ch$ |
| ResBlock up $4ch \to 2ch$ | ResBlock $16ch \to 16ch$ |
| BN, ReLU, $3 \times 3$ Conv $2ch \to 3$ | ReLU, Global average pooling ($f_x$) |
| Tanh | Linear $\to 1$ ($s_x$) |

## APPENDIX D    ADDITIONAL RESULTS OF CAUSAL CONTROLLABLE GENERATION

In this section we present more qualitative results of causal controllable generation on two datasets using DEAR and baseline methods, including S-VAEs (Locatello et al., 2020b) and CausalGAN (Kocaoglu et al., 2018). We consider three underlying structures on two datasets: Pendulum in Figure 2(a), CelebA-Smile in Figure 2(b), and CelebA-Attractive in Figure 2(c).

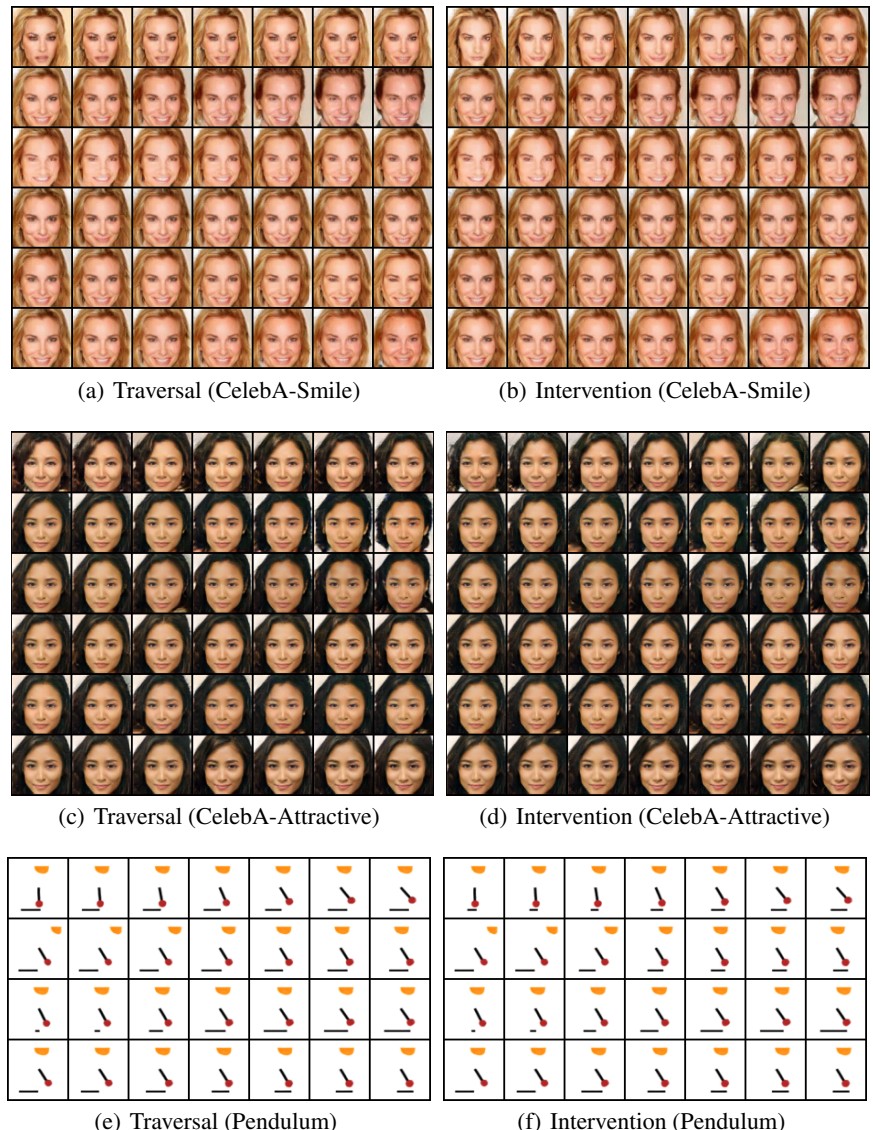

(a) Traversal (CelebA-Smile)  (b) Intervention (CelebA-Smile)

(c) Traversal (CelebA-Attractive)  (d) Intervention (CelebA-Attractive)

(e) Traversal (Pendulum)  (f) Intervention (Pendulum)

Figure 8: Results of DEAR. Note that the ordering of the representations matches that of the indices in Figure 2. On the left we show the traditional latent traversals (the first type of intervention stated in Section 5.1). On the right we show the results of intervening on one latent variable from which we see the consequent changes of the others (the first type of intervention). Specifically intervening on the cause variable influences the effect variables while intervening on effect variables makes no difference to the causes.

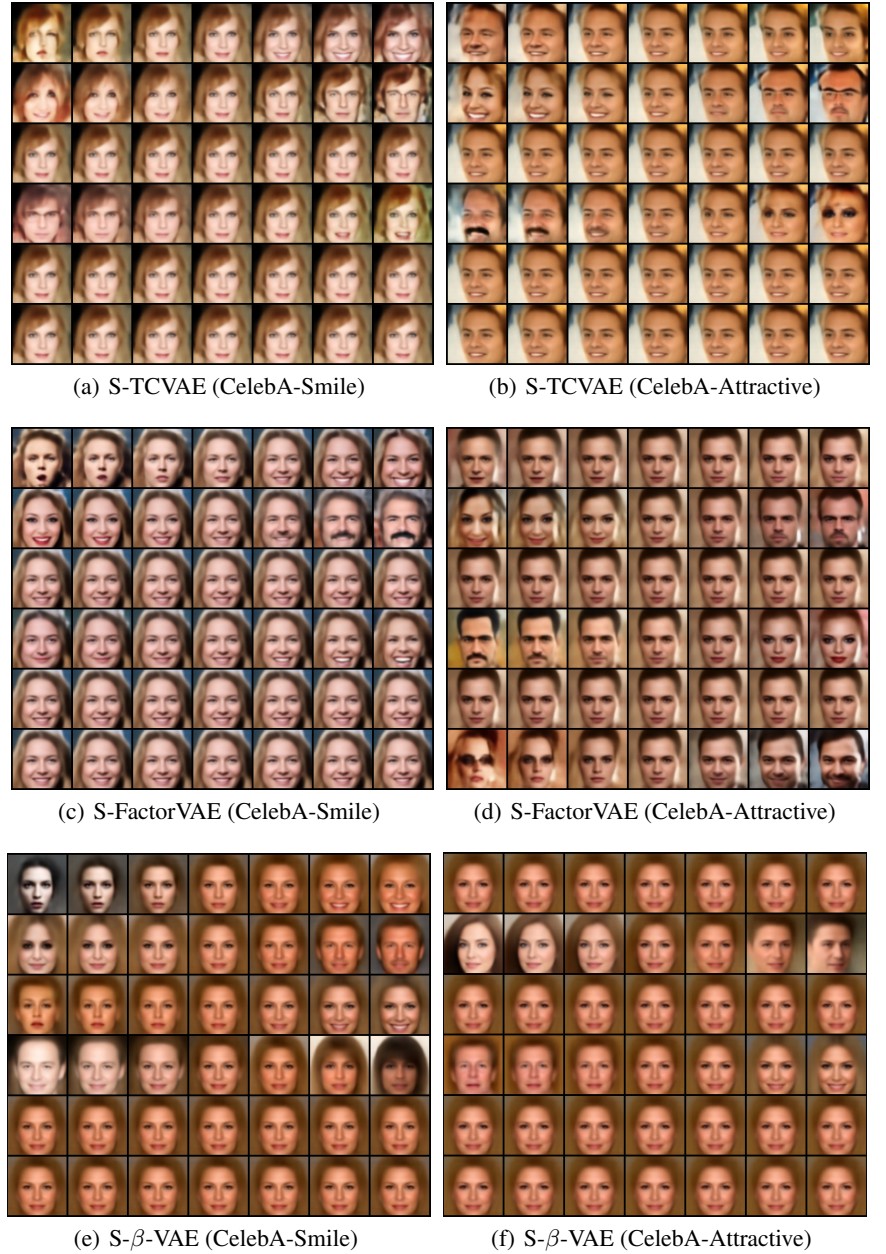

(a) S-TCVAE (CelebA-Smile)  (b) S-TCVAE (CelebA-Attractive)

(c) S-FactorVAE (CelebA-Smile)  (d) S-FactorVAE (CelebA-Attractive)

(e) S-$\beta$-VAE (CelebA-Smile)  (f) S-$\beta$-VAE (CelebA-Attractive)

Figure 9: Traversal results of baseline methods. We see that (1) entanglement occurs; (2) some factors are not detected (traversing on some dimensions of the latent vector makes no difference in the decoded images.) Besides, the generated images from VAEs are blurry.

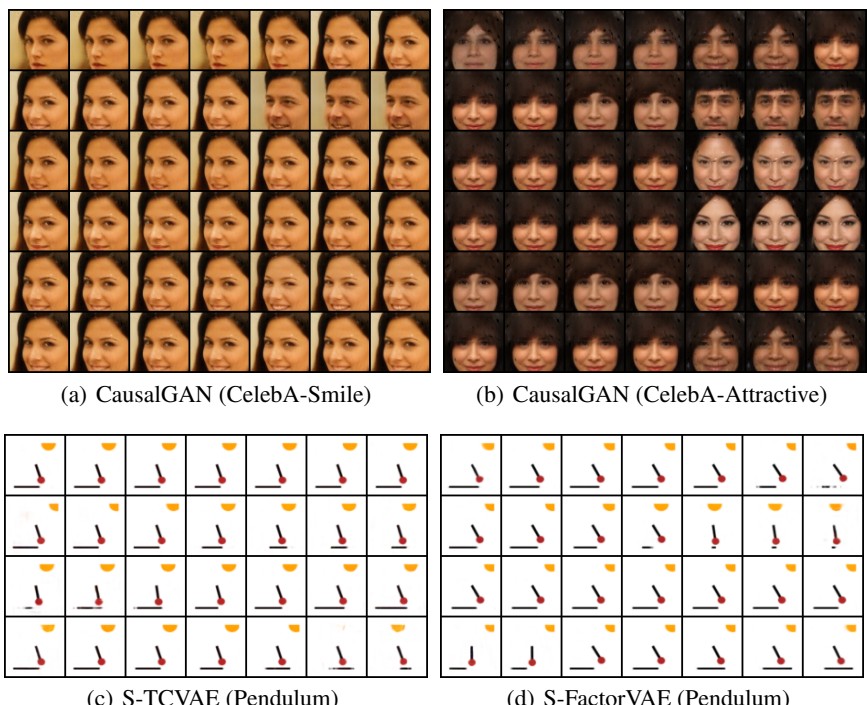

(a) CausalGAN (CelebA-Smile)    (b) CausalGAN (CelebA-Attractive)

(c) S-TCVAE (Pendulum)    (d) S-FactorVAE (Pendulum)

Figure 10: Traversal results of baseline methods. CausalGAN uses the binary binary factors as the conditional attributes, so the traversals appear some sudden changes. In contrast, we regard the logit of binary labels as the underlying factors and hence enjoy smooth manipulations. In addition, the controllability of CausalGAN is also limited, since entanglement still exists. Results of S-VAEs are explained in Figure 9.

