# OpenReview forum: "Disentangled Generative Causal Representation Learning"
_ICLR.cc/2021/Conference — Reject_

### Official Review · AnonReviewer2 · 2020-10-27
**Very interesting topic, but too many technical flaws, inaccuracies, and missing details in the current form**

**Rating:** 5
**Confidence:** 4

**Review:**

**Summary of the paper**
This submission addresses the topic of causal representation learning/causal disentanglement, i.e., learning explanatory latent factors which are not mutually independent, but instead connected via an underlying causal model. The authors propose the DEAR framework to address this task. DEAR consists of an encoder-decoder architecture (similar to VAEs) which is combined with a separate discriminator. The key learning procedure is to match two joint distributions over data x and latents z: (i) the joint induced by pushing the (empirical?) data distribution q(x) through the encoder q(z|x); and (ii) the joint induced by pushing the (model?) prior p(z) through the generator p(x|z). The task of distribution matching is done using a separate discriminator trained to distinguish q(x,z) from p(x,z) while encoder and decoder are trained to fool the discriminator.
Another key ingredient is the form of the model prior p(z) which does not factorise, but instead takes a particular form inspired by structural causal models (SCMs).
In order to learn the dependent latent variables, two additional strong assumptions are made: (i) full supervision in the form of observations of all ground truth causal factors y for (a subset of) the training data; (ii) ground truth knowledge of the causal ordering or full causal graph. The latter (ii) is used to construct the prior p(z) to allow for the same DAG structure as the true factors, and the former (i) is used to augment the joint distribution matching objective with a supervised regulariser that penalises deviation of the encoded factors from the groundtruth factors.
The authors claim to prove non-identifiability for models assuming a factorised prior, as well as identifiability for their model, but I have some reservations and concerns about the level of detail and technical correctness of the proofs (more details below).
Experiments are performed on three datasets with causally related latent variables: Pendulum, CelebA-Smile and CelebA-Attractive. The proposed DEAR approach is compared against a baseline (S-$\beta$-VAE) in terms of latent traversals, controllable generation, and two downstream tasks (sample efficiency and distributional robustness) and outperforms the baseline on these.

**Summary of evaluation**
While the paper addresses a very important problem and contributes some interesting ideas and results, the submission has too many technical flaws and misses too many important details to merit publication in the current form.

**Pros**
- the setting of learning causally related latent factors considered in the paper is a very interesting and highly relevant
- the approach of matching the two joint distributions induced by encoder and decoder in an adversarial fashion is well-motivated, interesting, and (as far as I can tell) novel
- the authors (attempt to) provide some theoretical insights along with the proposed method which is appreciated
- the datasets seem suitable for the task, and some of the results (e.g., Fig.3) are quite interesting and promising

**Cons**
- the main part of the paper (Sections 3 & 4) is quite hard to parse due to inconsistencies, non-standard use of notation (CE, q(x) for data distribution, ...), and important missing details (e.g., the assumed data generating process in terms of $\xi$ is never specified)
- the model relies on known ground truth causal ordering AND supervision in the form of annotated ground truth factors which is a very strong assumption and too unrealistic to be useful for causal representation learning in practice: learning the causal variables and their structure is precisely the task in causal disentanglement and it seems that most information is assumed given. At the same time, the importance of these assumptions is not stressed enough in the abstract and introduction and formulations like "using a suitable GAN loss" are misleading since they suggest an unsupervised approach.
- only linear relationships between the causal variables $V=f^{-1}(Z)$ with additive noise are allowed (before a component-wise nonlinearity f is applied) and it is not clear why this restriction is necessary or why the component wise nonlinearity f cannot simply by absorbed into the decoder
- the theoretical results claimed in the paper are not backed up by rigorous proofs (I checked proofs for Proposition 1 and Theorem 1, but not for the derivations of the gradients/Lemmas): there are several mathematical errors, missing assumptions, and unclear steps brushed under the carpet
- there is no information about the choice of two key quantities, the hyperparameter $\lambda$ determining the weight of the supervised regulariser, and the fraction/amount of labelled examples $N_S$ (nor any ablation on how performance depends on these) which harms transparency and reproducibility

**Comments regarding the Proof of Proposition 1**
- 2nd sentence: E should be E*.
- The assumption of a non-factorising distribution over $\xi$ is not stated as part of the proposition. This should be stated as a clear assumption in which case the first paragraph of the proof becomes obsolete.
- 2nd paragraph, 2nd sentence refers to distributions of z and E(x), but I believe E*(x) is meant. Otherwise the claim that "the intersection of their distribution families" (could also be clarified) is incorrect or at least not justified.
- 3rd paragraph refers to $p_E(x,z)$ which is not defined. I assume $q_E(x,z)$ is meant?
- A key problem I see with the definition of b as the minimum of $L_{sup}$ subject to the constraint $L_gen=0$. It is not clear why this minimum should exist, i.e., why it should be possible to satisfy the constraint if the prior $p(z)$ factorises, but the true generative process is based on dependent latents. Certainly, there are not sufficient details provided to assess the correctness of the statement.
- there are missing $\lambda$'s in $L^*$, should be $L^*$ $\geq a+\lambda b^* > \lambda b^*$

**Comments regarding the proof of Theorem 1**
- as stated before, cross entropy is between two distributions does not involve a sigmoid function
- what is the distribution $p(y_i|x)$? this was not defined in the paper
- the partial derivative is incorrect, should be $y/\sigma(E_i)-(1-y_i)/(1-\sigma(E_i))$ inside the bracket
- the second partial derivative misses a factor 2
- the main part of the proof is only informally sketched in natural language: this is not sufficiently rigorous for a prove that claims to show identifiability; more detail is needed to be able to assess the correctness of the claim. It is also not clear whether infinite data is assumed.

**More detailed comments and questions**
- the use of $q_x(x)$ for the data distribution is unconventional: usually this is denoted $p_{\text{data}}(x)$ or simply $p(x)$; also it would be good to distinguish between the empirical distribution provided by the training data and the true distribution which is unknown at different parts in the paper
- the related work section is very short and key references are not described in any detail: it would make the paper more accessible if key building blocks of the proposed method (e.g., Locatello et al., Yu et al.) were explained in more detail here
- I do not follow the logic  of the first sentence at the top of page 3, please clarify or rephrase
- the assumed true data generating process to be captured is never specified: in particular, what is the generative process for the ground truth factors $\xi$ and how are these factors decoded to give rise to the observations?
- I do not understand the need to consider separate $\xi$ and $y$? what does the addition of $y$ add? also what distribution is the expectation in the definition of $y$ with respect to: $\xi_i=\mathbb{E}[y_i|x]$ ?
- the definition of cross entropy stated in 3.2 does not coincide with the universally accepted form of cross entropy (which does not have the sigmoid term)
- what is $\phi$ in $L_{sup}(\phi)$ in 3.2? it is not defined at this point, and why does it not appear as argument to the other $L_{sup}$'s?
- Definition 1 seems strange to me as it does not allow for permutations of the ground truth factors. I would have expected that for all i there is a j s.t. $E_i(x)=g_j(\xi_j)$. Can you elaborate?
- last paragraph in Section 3: The claim made here seems incorrect: (E',G') as constructed in Proposition 1 minimises neither (1) nor (2), but $L_{sup}(E)$ subject to $L_{gen}=0$; see also my point on the feasibility of this constraint in the context of the proof of proposition 1
-Sec.4.1: if f is a component wise tranformation, it is not clear what it's addition adds, and why you do not consider a causal model over $V=f^{-1}(z)$ instead. the particular choice of a linear SCM is not motivated further which makes it hard to understand. Also, it would help to specify $p(\epsilon)$ earlier as otherwise one wonders what the point of $h$ is.
- you refer to the "direction" and scale of causal effects, but the direction is given by assumption in the form of causal order. do you mean the "sign" of the linear effect?
- the procedure described for computing interventions is not motivated from the latent SCM, and in fact it appears to be closer in nature to a counterfactual (rung 3) rather than an interventional (rung 2) quantity since the noise variables are kept fixed when performing the intervention as far as I understand. For an intervention, the exogenous variables would have to be re-sampled.
- I do not understand the reasoning in the last paragraph of Sec. 4.1.: the claim that k=m fails due to capacity issues is not explained and does not make sense to me. I do not understand why one would consider different sets of latent, since independent factors can also be incorporated within the latent SCM, these would simply have zeros in the corresponding rows and columns in the adjacency matrix.
- Theorem 1 assumes that "the true binary adjacency matrix" can be learned. This is too vague: what does "can be learnt" mean exactly? Also, why is the adjacency matrix binary all of the sudden?
- 4.3 states that unlike prior work, the prior distribution has learnable parameters, but there are a number of works considering learning the prior. you should either rephrase or reference such works.
- Algorithm 1: $\psi$ is not defined at this point. I assume it refers to the discriminator parameters?
- 5.1 you state that you can only manipulate the causal factors whilr leaving their effects unchanged, but this is incorrect. Effects will change unless you also intervene on those and fix them (which I believe is what you mean, given the experiments).
- There is a reference to Figure 3-9 which confused me.
- 5.2 I find the choice of defining "attractiveness as a slim young woman with makeup and thick hair" very questionable from an ethical perspective. I think that we should try our best in academia to overcome such outdated notions, and given that this was a design choice you made, I think it would be nice to consider a more neutral/less controversial example. (This is just a personal note and has no bearing on my evaluation!)

**Post rebuttal comments**
I thank the authors for the detailed response. I think that some points have been clarified and corrected, and I have increased my score slightly to reflect this. I still think the paper needs another iteration before publication though.

---

> ### Author Response · Authors · 2020-11-20
> **Addressing the cons**
>
> Thanks a lot for the detailed comments and suggestions. We address your concerns one by one as follows.
>
> * There are no inconsistencies in Sections 3 \& 4. All the notations are well-defined: specifically here we explain CE in point 1 of "Regarding the proof of Theorem 1" and $q(x)$ in point 1 of "We address the detailed comments and questions one by one".
> We agree with you that it is more complete to include the assumption on the data generative process and add them in the updated version. Specifically we follow the commonly assumed data generating process that first samples the underlying generative factors, and then conditional on those factors, generates the data. We stated before that $\xi$ follows a causal prior (i.e., connect by a causal graph) and now make it more explicit by introducing a notation $p_\xi$, which is assumed to be contained by the distribution family induced by the SCM (3), as formally described in Assumption 1, and is identifiable as stated in Assumption 2.
> Under the modified explicit definition and assumptions, we make clearer the required conditions in Theorem 1 to achieve identifiability of disentanglement.
> * It should be noted that currently strong enough supervision is required to achieve causal disentanglement with theoretical guarantee. 1) For the true causal ordering, in many applications it is available because the humans usually have some prior knowledge on high-level causal variables based on common sense and intuition. As mentioned in the paragraph after Fig. 1, the focus of this paper is disentanglement whose goal lies in the alignment between latent variables and the causal factors, while causal discovery is a separate topic on which one needs to explore detailedly the identifiability of the causal structure itself. To avoid too much separate discussion, we assume a necessary condition to ensure that the model learns the true causal structure. 2) Incorporating annotated ground-truth factors are common settings considered in recent disentanglement methods, as said in paragraph 2 in Sec 1.
> Besides, we did clearly mention in introduction that we incorporate supervision. "GAN loss" commonly refers to the adversarial training involving a discriminator but is definitely not limited to unsupervised learning, e.g., conditional GANs.
> Below we explain what our method is able to learn far more than the provided information.
> 	* We learn the SCM including the transformations $f$ and the weighted adjacency matrix given the causal ordering, which enables one to generate the causal variables from the exogenous variables, that is, enabling a generative model to sample from the SCM prior.
> 	* The annotated factors only help the alignment of the latent variable and the factors, but our model learns the whole bidirectional generative model (BGM) that can infer from or manipulate existing training/test data and controllably generate new data. Besides, our model outperforms the ERM method with the same amount of annotated labels in two downstream tasks, as shown in Table 1 ResNet-pretrain and Table 2 ResNet-multi.
> * We adopt the SCM (3) proposed by Yu et al. (2019) as the latent prior, because it generalizes a linear SCM, and allows both a generative form (3) to enable generation and a structural equation form (4) to enable intervention. However we do not mean only a linear additive form (after nonlinearity) is allowed. Actually any other choices that allow both a generative form and a structural equation form can be applied. The adopted one here is just one choice borrowed from a recent work in causal discovery that turns out to work well in the experiments.
> As to the nonlinearity, if $\xi$ appears non-linear relationships, then by minimizing loss $L$, the SCM prior will learn the nonlinearity $f$ to match it, and hence will not be absorbed into the decoder.
> * As to theoretical results, we address all the concerns one by one below.
> * We find the choice of $\lambda$ quite insensitive to different datasets and tasks, and hence set $\lambda=5$ in all experiments. Unless otherwise specified, the results shown in Section 5 use the full sample supervision, i.e., $N_s=N$. We also conduct experiments with $N_s=0.1N$. For controllable generation in Section 5.1, the qualitative results with 10% labels have no big difference with those with a full sample, so we only show the latter. For downstream tasks in Section 5.2, we report the quantitative results with 10% labels, as shown in Table 1-2 (DEAR-lin/nlr-10%) and give a discussion as in the end of Sec 5.2.1. We further add the related information in the revised paper. To help transparency and reproducibility, we attach the source code in Supplementary Materials.

---

> > ### Author Response · Authors · 2020-11-20
> > **Regarding the proof of Proposition 1 and Theorem 1**
> >
> > **Regarding the proof of Proposition 1**
> >
> > * Typos fixed: $E$ should be $E^*$ in the first two paragraphs, $p_E(x,z)$ should be $q_E(x,z)$, and the missing $\lambda$'s are added in the last paragraph.
> > * First, we state the assumption that the elements of $\xi$ are connected by a causal graph whose adjacency matrix is not a zero matrix in the paragraph before Prop 1, which we now move it into the proposition statement in the updated version. This assumption easily implies the non-factorized property of $\xi$ as described in the proof. However, since throughout the paper we are assuming the causal correlation within factors $\xi$, we think it is better not to state the non-factorized assumption but the causal correlation assumption to be consistent. Second, even with the non-factorized assumption on $\xi$, we still need the non-factorized property of $E^*(x)$ from Definition 1. Thus the first paragraph is not entirely obsolete.
> > * "Infeasible constraint": It is worth pointing out that in generative models, we first assign a latent prior $p(z)$ and then learn an encoder such that the distribution of its output matches $p(z)$. Without any additional regularizations concerning the ground-truth generative factors $\xi$, the learned BGM has nothing to do with $\xi$, because the latent variables in the model need not to align with $\xi$. Thus the fact that $\xi$'s are marginally dependent does not mean that we cannot learn a BGM with an independent prior such that $L_{gen}=0$. Instead, for example, learning under such a formulation may return a BGM whose latent variables $z$ are the independent components of $\xi$. If this is the case, it can be seen that the supervised loss will never be optimized. That is to say, with an independent prior, the generative modeling loss and the supervised regularizer contradicts between each other. In proposition 1, we give the condition (mainly on $\lambda$) under which the objective value $L$ at a BGM with an independent prior satisfying $L_{gen}=0$ is smaller than that at a BGM with a disentangled encoder.
> >
> > **Regarding the proof of Theorem 1**
> >
> > * The cross-entropy loss given after (2) is a quite standard form for binary classification, where sigmoid is used to transform the logit $l$ into the probability (within $(0,1)$). In other cases people may absorb sigmoid into $l$, e.g., as the last activation layer of a neural net. Here we just explicitly write it out, which is also used in [1].
> > * This is the case where $y_i$ is the binary label of $\xi_i$, so according to the basic property of a Bernoulli distribution we know $P(y_i=1|x)$ $=$ $\mathbb{E}(y_i|x)$ and $P(y_i=0|x)=1-\mathbb{E}(y_i|x)$, which characterize the probability mass function $p(y_i|x)$.
> > * Ours is correct. By some simple algebra we know these two expressions are identical, i.e,  $y/\sigma-(1-y)/(1-\sigma)=y/(\sigma(1-\sigma))-1/(1-\sigma)$ for $y:=y_i$ and $\sigma=\sigma(E_i)$.
> > * Added the factors of 2 in the new version.
> > * In Section 4.2, we propose the formulation (6) at the population level, meaning that infinite data is assumed. We then establish the identifiability of formulation (6) which is again stated at the population level. In the updated paper, we explicitly state Assumption 1-2 on the SCM capacity and the identifiability of the true structure, based on which we revise the last part of the proof.

---

> > > ### Author Response · Authors · 2020-11-20
> > > **Addressing the detailed comments and questions one by one.**
> > >
> > > * [Notation $q_x$] In the encoder-decoder literature, it is conventional to use $q$ to denote the distributions associated with the encoder and $p$ to denote those with the decoder/generator. To make notations in our paper consistent, we use $q$ to denote the probability functions involved in the inference process (from the real data to the inferred latent), and use $p$ to denote those involved in the generation process (from the latent prior to the generated data). We clearly defined $q_x$ at the very beginning not to bring confusions. Besides there are literature using $q$ to denote the real data distribution in the field of generative models like [2].
> > > * [More discussion on related work] We add the related discussion on Locatello et al. in Section 3.2 and that on Yu et al. in Section 4.1, which we agree makes the paper more accessible.
> > > * [First sentence at the top of page 3] In a word, here we try to explain the necessity of more general distributions beyond factorized Gaussians for $q_E$ and $p_G$. We point out two reasons. First, to achieve disentanglement of a BGM, we are essentially enforce some constraints on the latent space, so the distribution family of $q_E$ and $p_G$ should be large enough to satisfy these constraints. This corresponds to the infinite capacity assumption in Theorem 1. Second, to model complex real data like images, we need more general generated conditionals like implicit distributions, as shown in the literature (Karras et al., 2019; Mescheder et al., 2017).
> > > * [True data generative process] See point 1 of "Addressing the cons".
> > > * [$\xi$ and $y$] $\xi$ denotes the ground-truth generative factors of $x$ while $y$ is the observation of a discrete or continuous form satisfying $\xi_i=\mathbb{E}(y_i|x)$ where the expectation is taken wrt the underlying conditional distribution $p(y_i|x)$. For example, in the case of human face images, $y_1$ can be the binary label "young" ($y_1=1$ refers to young) and $\xi_1=\mathbb{E}(y_1|x)=P(y_1=1|x)$ is the probability of being young given one face image $x$.
> > > * The definition of cross entropy loss is discussed in point 1 of our response regarding the proof of Theorem 1.
> > > * The redundant $\phi$ in $L_{sup}(\phi)$ in 3.1 is a typo that is removed in the new version.
> > > * [Definition 1] In general the goal of disentanglement allows for permutations, but here since in our method we supervise each latent dimension by each annotated label of the ground truth factor, we can expect an component-wise correspondence between $E(x)$ and $\xi$, i.e., $E_i(x)=g_i(\xi_i)$ for the dimension $i$. Thus this definition serves for the analysis and justification of our proposed method, which should make sense.
> > > * [Last paragraph in Section 3] Please note that the reference to (1) should be (2), apart from which the claim is correct. We do not say $(E',G')$ minimizes (2) but the value of objective (2) evaluated at $(E',G')$ is smaller than that at any disentangled solution $(E^*,G)$ under the general condition stated in Proposition. Hence by minimizing (2), we have no way to end up with a disentangled solution $(E^*,G)$.
> > > * [Choice of SCM] As mentioned after equation (4), the adopted SCM indicates $z$ satisfies a linear SCM after nonlinear transformation $f$, so one can view it as an SCM over $V=f^{-1}(z)$. However $f$ needs to be learned, so we need to involve them in the model. The motivation of SCM is re-emphasized above in point 3 of "Addressing the cons". We specify $p_\epsilon$ after equation (3) in the new version.
> > > * Yes it refers to the sign rather than the causal direction. We have revised it to avoid confusions.
> > > * [Intervention procedure] Actually we follow the formalization of intervention as operations that modify a subset of equations in (4) (Pearl, et al., 2000), which is exactly motivated from the SCM. Note that such operations can modify either the whole equation or the exogenous variables $\epsilon$, so $\epsilon$ can be either fixed or resampled for an intervention. In addition, we would like to clarify that in our procedure, the noise variables are not necessarily kept fixed, but they can be resampled and the consequent traversals remains similar.

---

> > > > ### Author Response · Authors · 2020-11-20
> > > > **Addressing the detailed comments and questions one by one (cont.)**
> > > >
> > > > * As to the last paragraph of Sec. 4.1, first please note that $m$ denotes the number of factors of interests which can be much smaller than the total number $M$ of generative factors of data. The latent dimension $k$ of the generative model should be no less than $M$ to allow a sufficient degree of freedom in order to generate or reconstruct data well. Since $M$ is generally unknown in reality, the idea here is to set a sufficiently large $k$ (at least larger than $m$ which is a trivial lower bound of $M$).
> > > > For the first set of latent variables of dimension $m$, we explicitly use an SCM prior. The other $k-m$ dimensions are used to capture other factors necessary for generation, for which we can assign any prior distribution because we do not care about the structural dependence among them. Only under the special case where we use a factorized Gaussian prior, we can incorporate them within the SCM and expand the adjacency matrix using zero blocks. So the statements in the paper are more general, which is further modified to be clearer in the updated version.
> > > > * [Assumptions in Theorem 1] We rephrase the assumption of the learnability of the true binary adjacency matrix by Assumption 1-2 which state the SCM capacity and the identifiability of the true structure. Besides, the binary adjacency matrix refers to the binary version of the weighted one. As said in the paragraph after Figure 1, given the binary adjacency, we learn the weights of the non-zero elements. So the adjacency matrix in the model to be learned is still weighted, but the prior knowledge is given on the binary adjacency.
> > > > * [Previous work with learnable priors] We refer to the previous works using the generative modeling objective (1) without a closed-form, like in most GAN methods, which do not learn the prior, but use $N(0,I)$. Hence the gradient formulas presented in Lemma 1 are necessary. We rephrase the description to "different from Shen et al., (2020)".
> > > > * In Algorithm 1, $\psi$ denotes the parameter of the discriminator. Added in  the new version.
> > > > * In paragraph 2 of Sec. 5.1, we describe the two types of interventions that we conduct. Specifically the one you concerned with here is the first one which fixes all the latent variables and varies one. When the one corresponds to the cause variable, the outcome in the traversals is that we only manipulate the cause factors while leaving their effects unchanged.
> > > > * Fig. 3-9 ---> Fig. 3-4. Sorry for the incorrect reference.
> > > > * [Ethical issue] Thanks for pointing this out. We agree and change the definition of the target label to ONE KIND of attractiveness and emphasize in the paper that the definition of attractiveness here only refers to one kind of attractiveness, which has nothing to do with the linguistic definition of attractiveness.
> > > >
> > > >
> > > > [1] Locatello, F., Tschannen, M., Bauer, S., Rätsch, G., Schölkopf, B., & Bachem, O. (2020). Disentangling Factors of Variation Using Few Labels. In ICLR, 2020.
> > > >
> > > > [2] Dumoulin, V., Belghazi, I., Poole, B., Lamb, A., Arjovsky, M., Mastropietro, O., & Courville, A. C. (2017). Adversarially learned inference. In ICLR, 2017.

---

### Official Review · AnonReviewer1 · 2020-10-29
**An interesting paper on generative latent space modelling with SCMs, more discussion on relation to similar models needed.**

**Rating:** 6
**Confidence:** 3

**Review:**

This paper presents a latent variable model where the variables in the latent space are causally disentangled, i.e. the disentanglement is ensured according to a structural causal model (SCM). The resulting model is made up of two parts. The first one, a generative unsupervised part, is essentially a VAE and is defined with the VAE ELBO loss. The second part is supervised and accounts for the causal disentanglement of the factors that are assumed to underlie the distribution; the authors claim the fewer supervised samples are required to estimate the second part of the loss alone. The two parts are then combined with a hyperparameter.

The authors motivate their approach by comparing it to models using an independent prior for the latent factors and to ones using structural causal models in the latent space. For the former case, they claim that in reality the latent factors are frequently dependent, which their model is able to capture. For the latter, they argue that the competing models use SCMs for conditional factors rather than unconditional as in the proposed approach.

Subsequently, disentangled representation is defined as one where a 1-to-1 function can be established between the data and the underlying factors corresponding to the nodes of the SCM (Definition 1). The authors then propose to use the general nonlinear SCM model in the latent space which concludes the actual definition of their model (part 2 of the loss mentioned before). Finally, they note that the model ensures disentanglement as defined before and describe a GAN-like algorithm for minimizing the compound loss (part 1 + part 2). The paper is concluded with a series of experiments which include a quantitative comparison of accuracy, sample efficiency and distributional robustness against a number of VAE-based disentanglement methods.


*****Strengths:*****

The incorporation of structural causal models to generative modelling, especially in the context of disentanglement and the resulting explainability, is an important topic and this paper is certainly relevant in this area.

The experiments show a quantitative improvement over a number of competing disentanglement methods.


*****Weaknesses:*****

Generative modelling with SCMs, being an important area of research, was addressed before. The review of related work provided here is rather limited. Moreover, the authors decided to contrast their approach mostly to disentanglement strategies with an independence prior. I think the paper would be much more convincing if more weight were put on direct comparison with other methods using SCMs for generative modelling such as causalGAN and cognate methods (which is absent from the experiment section at the moment). As a matter of fact even the limited high-level comparison to such models provided in section 2 left me confused (what is the “unidirectional nature” of the other models compared to DARE? What is meant by “direct access to attributes” in the other methods – labels? Factors?).


*****Questions / feedback:*****

See weaknesses.

What is the role of the hyperparameter lambda (equations 2, 6)? Can we alter the sample complexity (wrt labels y) by varying it, or make the model less dependent on the assumed generalized SCM?

Why is there a difference in notation between Z and \xi? Is the DAG defined by Z not the same as the SCM defined by \xi?


*****Typos / minor comments:*****

Section 4.2: justification on -> justification of

Section 4.1: interventional distribution is never defined

I found Figure 1 unreadable.


*****Post Rebuttal*****

I would like to thank the authors for the rebuttal. While I think more discussion / comparison to approaches without an explicit independence prior would further improve the paper, the authors have clarified many of my doubts. I have therefore decided to raise my final score.

---

> ### Author Response · Authors · 2020-11-20
> **Response to Reviewer1**
>
> Thanks for the comments and suggestions. We address your concerns as follows.
>
> 1) [CausalGAN] First, CausalGANs are unidirectional generative models, meaning that they only learn a mapping from latent to data without an encoder in a reverse direction, so they cannot learn representations (i.e., infer latent variables) from data or manipulate a given data (like what we did in Sec. 5.1 and 5.2). This is the reason why CausalGANs are not regarded as the main baseline methods. About "direct access to attributes", attributes here refer to the annotated factors. We modify this sentence into "the ground-truth factors are directly fed into the generator as the conditional attributes, without an extra effort to align the dimensions between the latent variables and the underlying factors, so their models have nothing to do with disentanglement learning." We also add more detailed explanations on the limitations of CausalGANs compared with ours in the revised paper, especially for people who may not be familiar with the setups in generative models.
> Second, CausalGANs can do controllable generation limited to the settings to generate new data (but not manipulate given data), and with binary generative factors but not continuous ones. We present the results of CausalGAN in Fig. 10 of Appendix D which appear non-smooth with sudden changes and limited controllability.
>
> 2) The role of the hyperparameter $\lambda$ is to balance the generative model loss and the supervised loss, leading to balanced optimization of both losses. Empirically we find the choice of $\lambda$ quite insensitive to different datasets and tasks, and hence set $\lambda=5$ in all experiments.
> By varying it we indeed can alter the sample complexity wrt $y$. A formal derivation of the sample complexity is an interesting topic worth exploring in future work.
>
> 3) [Difference between $\xi$ and $z$] $\xi$ is the underlying ground-truth factors, and $z$ is the latent variable of our generative model (a model element). One can specify a prior for $z$ and learn the generative model by minimizing $L_{gen}$, which has nothing to do with $\xi$ unless we involve supervision on $\xi$. The case of $z=\xi$ corresponds to one specific generative model. As defined in Definition 1, we pursue a 1-1 correspondence between each dimension of $z$ and $\xi$, not necessarily an identity mapping. As to the DAG, the causal structure of $z$ should be the same as that of $\xi$, but the weighted adjacency matrix and nonlinear transformations can vary.
>
> 4) [Interventional distribution] We define the interventional distributions that we consider in experiments in paragraph 2 of Section 5, by defining how the intervention alters the structural equations in equation (4), which induces the interventional distribution.
>
> 5) We modify Fig. 1 in the updated version.

---

### Official Review · AnonReviewer4 · 2020-10-30
**Disentangled Generative Causal Representation Learning**

**Rating:** 6
**Confidence:** 4

**Review:**

The paper propose to learning causal disentangled representation which conforms human's cognition. The learned representation can be applied in causal controllable generation and benefit downstream tasks. The uses of this method is interesting.
The paper has some advantages:
It only requires part of sample is supervised, the previous methods need fully supervied information.
The previous method unreasonably force the latent factors to be independent. However, this method incorporate a SCM as prior distribution of latent representations. Therefore, the learned factors can be causally correlated which is consistent to practice.
The experiments are extensively conducted. The results demonstrate the effectiveness of this method.

I have some questions about this method:
In section 4.3, the gradient in Algorithm 1 is not consistent with the gradient in Lemma 1. Which one is correct?
In section 5.2.2, the test set is grouped into 4 groups according to the two binary labels, the target one and the spurious one. It is not clear what is the meaning of target one and spurious one.
The paper require the information about SCM as prior, which is a strong requirement. I think it is of great significance to investigate how to learn causal disentanlged representation with weaker prior knowledge in the future. I understant it is very difficult, but it is worth researching.

---

> ### Author Response · Authors · 2020-11-20
> **Response to Reviewer4**
>
> Thanks for the comments. Please find our responses below.
>
> 1) [Section 4.3] The gradient used in the algorithm is the estimation of that in Lemma 1, as mentioned in the paragraph after Lemma 1. Besides, in formula (7) we express the gradients explicitly by applying the chain rule while in algorithm we use a combined expression for simplicity.
>
> 2) [Section 5.2.2] The target one is the one which we predict (i.e., attractiveness on CelebA and corruption on Pendulum), while the spurious one is the label with the spurious correlation with the target label (i.e., mouth\_open on CelebA and background\_color on Pendulum).
>
> 3) [Weaker supervision] We agree that causal disentanglement under weaker prior knowledge is an important direction to explore. As to the two forms of supervision involved in our method: label supervision and causal structure, we conduct some ablation studies shown in Table 1-2 (DEAR-lin/nlr-10%) and Appendix B to suggest the potential of weakening both forms. More exploration on the exact amount of supervision needed to guarantee disentanglement is worth exploring in future work.

---

### Official Review · AnonReviewer3 · 2020-10-30
**This paper presents a novel and interesting method to formulate learning of correlated disentangled causal factors as a part of encode-decoder framework using a Gan-style learning approach. The proposed idea is interesting but I think the wording of the paper as well as the theoretical and experimental results could be improved.**

**Rating:** 5
**Confidence:** 5

**Review:**

This paper presents a novel and interesting method to formulate learning of correlated disentangled causal factors as a part of encode-decoder framework using a Gan-style learning approach. The proposed idea is interesting but I think the wording of the paper as well as the theoretical and experimental results could be improved.

Please find my detailed comments:
1- In equation (6), I suggest it to be modified to L(E, G, F)=L_{gen}(E,F,G)+\lambda L_{sup}(E). In other words, it would make sense to remove min_{E, G, F}.

2- Proposition 1: I think a necessary condition for a to be not equal to zero is that if the ground truth latent factors are correlated. Therefore, I suggest this assumption to be added to the assumptions of the proposition. Regarding the proof, I found the following typos. First, in the third line of the proof, I think x_{j} should be \xi_{j} instead. In the last two lines of the proof, L(E^*, G)\geq a+b^* should be replaced with L(E^*, G)\geq a+\lambda b^*.

3- Regarding theorem 1, I found the sentence "Then DEAR learns the disentangled encoder E*" not accurate. It is because assuming E, G and f does not guarantee that this framework is able to actually learn E*. Therefore, I suggest to change this statement to be more like the following statement: The encoder for the optimal solution of the proposed loss fn will be disentangled encoder E*. I found the following typos in the proof of this theorem. First, in the third line, inside the parenthesis, y should be replaced with y_i. In the 8th line, the derivative should be with respect to E_i(x) and not \sigma(E_i(x)). In the 11th line, p(x,z) should be replaced with p_G(x, z).

4- In the section 4.3 Algorithm, I found the following typos. First, in the second line, "implicit generated conditional" should be "implicit generative conditional". In the 5th line, "theorem" should be replaced with "Lemma". In Lemma 1, in the presented gradients, L_{gen} should be L.

5- Regarding the Experiments section, I did not find the qualitative experiments to clearly present the superiority of this method to disentangle correlated causal factors over the previous methods with a similar objective.

---

> ### Author Response · Authors · 2020-11-20
> **Response to Reviewer3**
>
> Thanks for the comments and for pointing out the typos which we have fixed in the revised version. We address your concerns as follows.
>
> 1- In (6), we present the learning formulation which is an optimization problem. We add $:=$ in the updated version.
>
> 2- We move the assumption into Prop 1 that "the elements of $\xi$ are connected by a causal graph whose adjacency matrix is not a zero matrix" stated in the paragraph before Prop 1.
>
> 3- Since DEAR formulation is given by (6), i.e., minimizing the objective $L$, the current statement "DEAR learns the disentangled encoder $E^*$" is identical to the one you suggested. We further clarify this statement by modifying into "DEAR formulation (6) learns the disentangled encoder $E^*$".
>
> 4- First, we call it generated conditional in line 5 of Sec 3.1 as an analogy of the encoded conditional. It is a convention in generative models literature to call the marginal distribution of the generated data "the generative distribution", but not necessary for a conditional distribution. Second, in Lemma 1, it should be $L_{gen}$ but not $L$. The gradient of $L_{sup}$ is straightforward while that of $L_{gen}$ is hard, so we present the gradients of $L_{gen}$ in Lemma 1.
>
> 5- We believe S-VAEs are previous methods with a similar objective (i.e., VAE + supervised loss). We compare in Fig. 3-4 with the typical one S-$\beta$-VAE. Specifically in block (a) of Fig. 3-4, S-$\beta$-VAE fails to disentangle the causal factors because manipulating one latent dimension and keeping others fixed show changes in multiple factors. In contrast, blocks (b) show the good disentanglement performance of ours. In Appendix D, we provide further comparisons with other S-VAEs.

---

### Author Response · Authors · 2020-11-20
**Summary of changes**

We would like to give our sincere thanks to all reviewers for their valuable comments. To address them, we make changes to our paper, which are summarized below. Please kindly refer to the updated version.
* For related work, we add more detailed discussion on CausalGAN in Section 2, on Locatello et al. (2020b) in the last paragraph of Section 3.1, and on Yu et al. (2019) before Figure 1 in Section 4.1.
* We explicitly add the assumed data generating process: the general assumption is given in Section 3.1, and specific assumptions on the causal graph of $\xi$ are stated in Assumption 1-2.
* We move the assumption on the causal correlation of $\xi$ into Proposition 1.
* We rephrase the assumption in Theorem 1 by Assumption 1-2.
* Typos and unclear/improper wording are fixed.
* Figure 1 is modified to be clearer.

---

### Decision · Program_Chairs · 2021-01-07
**Final Decision**

**Decision:**

Reject

**Comment:**

This paper presents a method to formulate learning of causally disentangled representation as a part of the encode-decoder framework. Although the reviewers agree that the paper presents some interesting ideas, they feel the paper is not ready for publication yet.  In particular, I encourage the authors to take the feedback of reviewer R2 into account, which is quite detailed and provides substantive ways of improving the work. After all, I recommend rejection.